# The evolution and biological correlates of hand preferences in anthropoid primates

Kai R Caspar[1,2]*, Fabian Pallasdies[3], Larissa Mader[1], Heitor Sartorelli[4], Sabine Begall[1]

[1]Department of General Zoology, University of Duisburg-Essen, Essen, Germany; [2]Department of Game Management and Wildlife Biology, Faculty of Forestry and Wood Sciences, Czech University of Life Sciences, Praha, Czech Republic; [3]Institute for Theoretical Biology, Department of Biology, Humboldt-Universität zu Berlin, Berlin, Germany; [4]Independent researcher, São Paulo, Brazil

**Abstract** The evolution of human right-handedness has been intensively debated for decades. Manual lateralization patterns in non-human primates have the potential to elucidate evolutionary determinants of human handedness, but restricted species samples and inconsistent methodologies have so far limited comparative phylogenetic studies. By combining original data with published literature reports, we assembled data on hand preferences for standardized object manipulation in 1786 individuals from 38 species of anthropoid primates, including monkeys, apes, and humans. Based on that, we employ quantitative phylogenetic methods to test prevalent hypotheses on the roles of ecology, brain size, and tool use in primate handedness evolution. We confirm that human right-handedness represents an unparalleled extreme among anthropoids and found taxa displaying population-level handedness to be rare. Species-level direction of manual lateralization was largely uniform among non-human primates and did not strongly correlate with any of the selected biological predictors, nor with phylogeny. In contrast, we recovered highly variable patterns of hand preference strength, which show signatures of both ecology and phylogeny. In particular, terrestrial primates tend to display weaker hand preferences than arboreal species. These results challenge popular ideas on primate handedness evolution, including the postural origins hypothesis. Furthermore, they point to a potential adaptive benefit of disparate lateralization strength in primates, a measure of hand preference that has often been overlooked in the past. Finally, our data show that human lateralization patterns do not align with trends found among other anthropoids, suggesting that unique selective pressures gave rise to the unusual hand preferences of our species.

*For correspondence:
kai.caspar@uni-due.de

Competing interest: The authors declare that no competing interests exist.

## Editor's evaluation

This paper combines new and previously generated data on hand preference to show that hand preference strength, but not direction, is predicted by ecology and phylogeny across primates. By drawing on the most expansive data set to date on experimentally determined hand preference, it calls existing hypotheses on the evolution of hand preference into question and shows that the strength of lateralization in humans is unusually extreme. Its results are of interest for evolutionary anthropologists, primatologists, and morphologists interested in the evolution of lateralization and human uniqueness.

## Introduction

Pronounced right-handedness is a universal trait among extant human populations (***Coren and Porac, 1977***; ***Raymond and Pontier, 2004***; ***Faurie et al., 2005***) and might be an ancient attribute of the

**eLife digest** About 90% of humans are right-handed. While it is known that handedness is caused by certain brain regions that are specialized in one of the two hemispheres, it is not clear how this evolved or why right-handedness dominates. Several hypotheses have been proposed to explain this extreme preference, including the use of tools, the larger size of the human brain, and the fact that humans live primarily on the ground.

Many researchers have regarded the extreme population-wide preference for using the right hand as being uniquely human. However, handedness had not been studied in a standardized manner across a wide range of primates. To fill this gap in our knowledge and understand how handedness may have evolved in monkeys and apes, Caspar et al. used existing data and new experimental observations to create a large dataset of hand preference.

This dataset illustrates how approximately 1800 primates across 38 species retrieve mashed food from a tube (or pieces of paper in the case of humans). Similar to humans, some species of monkey only had small proportions of ambidextrous individuals. However, no species had an extreme preference for using one specific hand the way humans do. Interestingly, Caspar et al. found that the presence of tool use as well as brain size were not associated with the degree of handedness in species. However, ground-living primates tended to show weaker individual preferences for a specific hand than tree-living species, with humans being a notable exception to the trend.

These findings confirm that humans do exhibit exceptional right-handedness, being unique among primates. While the results cannot explain the cause of this behaviour, they do help to rule out some of the theories that aim to explain how this preference evolved. This will be of interest to researchers studying the origins of human behaviour as well as the emergence of asymmetries in the brain.

genus *Homo* (*Toth, 1985*; *Lozano et al., 2017*). Whereas a significant expression of manual lateralization at population level is not exclusive to humans, the universal proportion of approximately 85–95% right-handers in our species appears to be an unmatched extreme among primates (*Meguerditchian et al., 2013*). Furthermore, individual humans tend to be strongly handed with ambiguous hand preferences being extremely rare (*Cochet and Vauclair, 2012*), which is also unusual when compared to many other primate lineages (*Westergaard and Suomi, 1996*; *Hopkins et al., 2011*). Therefore, both strength and direction of population-level manual lateralization in humans are widely considered to be remarkable.

Handedness is a behavioral consequence of functional asymmetries in the brain (*Amunts et al., 1996*; *Häberling and Corballis, 2016*; *Sha et al., 2021*). Accordingly, right-handedness results from unilateral specializations in the left hemisphere and vice versa. In anthropoid primates, which encompass monkeys, apes, and humans, asymmetries of the precentral gyrus in the primary motor cortex show a particular association with hand preference at the individual level (*Yousry et al., 1997*; *Phillips and Sherwood, 2005*; *Dadda et al., 2006*; *Sha et al., 2021*). Nevertheless, the proximate reasons for the expression of individual manual lateralization in humans and other anthropoids, including its genetic basis and the influence of brain areas located outside of the motor cortex, are by no means fully understood (*Rogers, 2009*; *Hopkins et al., 2013a*; *Ocklenburg et al., 2014*; *Schmitz et al., 2017a*; *Richards et al., 2021*). For this paper, however, we exclusively focus on the evolutionary underpinnings of population-level hand preferences.

The origins of pronounced population-level right-handedness in the human lineage have traditionally been linked to the emergence of complex communication mediated by manual gestures and language, which are also predominantly processed in the left hemisphere (*Corballis, 1991*; *Annett, 2002*; *Meguerditchian et al., 2013*; *Ocklenburg et al., 2014*; *Prieur et al., 2019*). However, current evidence suggests that manual and language lateralization are not nearly as tightly correlated as once believed and functional ties between these phenomena remain unidentified (*Fitch and Braccini, 2013*; *Ocklenburg et al., 2014*; *Schmitz et al., 2017b*). It has also been established that various non-human primates show significant asymmetries in hand use at the individual and population level in a variety of tasks, including manual communicative gestures and bimanual actions (*MacNeilage, 2007*; *Hopkins et al., 2013a*; *Meguerditchian et al., 2013*; *Regaiolli et al., 2016*). Nevertheless, reports of significant population-level biases are largely confined to a comparatively small number of species

and the distribution of individuals across hand preference categories is always far more balanced than in humans. For instance, olive baboons (*Papio anubis*), Western gorillas (*Gorilla gorilla*), and chimpanzees (*Pan troglodytes*) have all been reported to show a significant population-level right-hand bias for bimanual manipulation but the portion of right-handers among these species is only around 50% (*Vauclair et al., 2005*; *Hopkins et al., 2011* – note that this skew is significant due to a notable portion of ambipreferent individuals among these species). Still, it is essential to discern what underlies these comparatively weak population-level hand preference patterns that emerged across the primate order to unravel the origins of pronounced right-handedness in our species (*MacNeilage, 2007*).

The most influential conjecture to explain how primate hand preference patterns evolved is the postural origins hypothesis (POH) (*MacNeilage et al., 1987*; *MacNeilage, 2007*). Considering galagos as models, the POH assumes that hypothetical insectivorous primate ancestors exhibited a right-hand bias to support their body on vertical substrates, while the left-hand specialized for fast grasping movements, the so-called ballistic reaching (*Ward, 1998*). Based on this, the POH predicts that with the emergence of anthropoid primates, which exhibit arboreal quadrupedalism and more refined digit control, the right hand became adopted to manipulate objects during foraging (*MacNeilage et al., 1987*). Hence, it proposes that all anthropoids share a right-hand bias for manipulation, which would find its most extreme expression in humans (*MacNeilage, 2007*; *MacNeilage et al., 1987*). In the anatomically less derived strepsirrhines, the left hand is instead expected to be dominant (*MacNeilage, 2007*). However, the POH has been drastically modified in more recent studies (*Hopkins et al., 2011*; *Meguerditchian et al., 2013*; *Morino et al., 2017*). The novel interpretation proposes (in conflict with the original POH) that arboreal monkeys and apes should display a left-hand bias for manipulation. Their right hand would provide the necessary postural support, retaining the hypothesized ancestral primate pattern laid out by *MacNeilage et al., 1987*. Terrestrial lineages, however, would no longer be bound to reserve the right hand for posture stabilization and are expected to evolve right-hand preferences for fine motor skills, eventually giving rise to the human condition. Hence, the novel POH expects that left-handedness is prevalent in arboreal groups, while a preponderance of right-handedness should be restricted to terrestrial primate species (*Meguerditchian et al., 2013*). In contrast, the original POH expects to find a right-handedness trend in all anthropoids, regardless of their ecology. Both versions of the POH exclusively rely on correlational evidence from behavioral studies on hand preferences in different primate groups.

In addition to these considerations, it has been prominently proposed that foraging-related extractive tool use facilitated the evolution of hand preferences in humans to allow for more efficient object handling (*Kimura, 1979*; *Frost, 1980*). By now, this idea has also been extended to non-human primates that habitually use tools (*Cashmore et al., 2008*; *Prieur et al., 2019*). Moreover, neuroanatomical studies demonstrated that the expression of overall neural lateralization and hemispheric independence positively correlates with absolute brain size in primates (*Rilling and Insel, 1999*; *Karolis et al., 2019*; *Ardesch et al., 2021*). Such a scaling relation was already hypothesized by *Ringo et al., 1994*, and implies that the strength of individual handedness could also be tied to absolute brain size (*Hopkins, 2013a*; the concept does, however, not concern the direction of hand preferences). Hence, it can be hypothesized that larger-brained primates should evolve greater manual lateralization strength to reduce the amount of interhemispheric communication needed to accomplish manipulative tasks.

Given all this, testable hypotheses on primate hand preference evolution have long been established, but none of them have so far been assessed within quantitative evolutionary frameworks (*Hopkins, 2013b*). Studies on comparative cognition increasingly rely on phylogenetically informed modelling to estimate how and when specific behaviors evolved (*MacLean et al., 2012*; *ManyPrimates et al., 2019*; *Krasheninnikova et al., 2020*). Such approaches can provide estimates of ancestral states and allow researchers to quantify the influence of phylogeny and ecological variables on cognitive evolution (*MacLean et al., 2012*). However, to yield meaningful results, dense taxonomic sampling of species and a consistent testing scheme are required (*Freckleton et al., 2002*; *Krasheninnikova et al., 2020*). For research on primate hand preferences, this means that subjects from different species need to engage in the same experimental task to assess lateralization. Standardization is particularly important since both the strength and the direction of individual hand preferences can be task-dependent and because the expression of manual lateralization often correlates positively with motor complexity (*Vauclair et al., 2005*; *Blois-Heulin et al., 2007*; *Lilak and Phillips,*

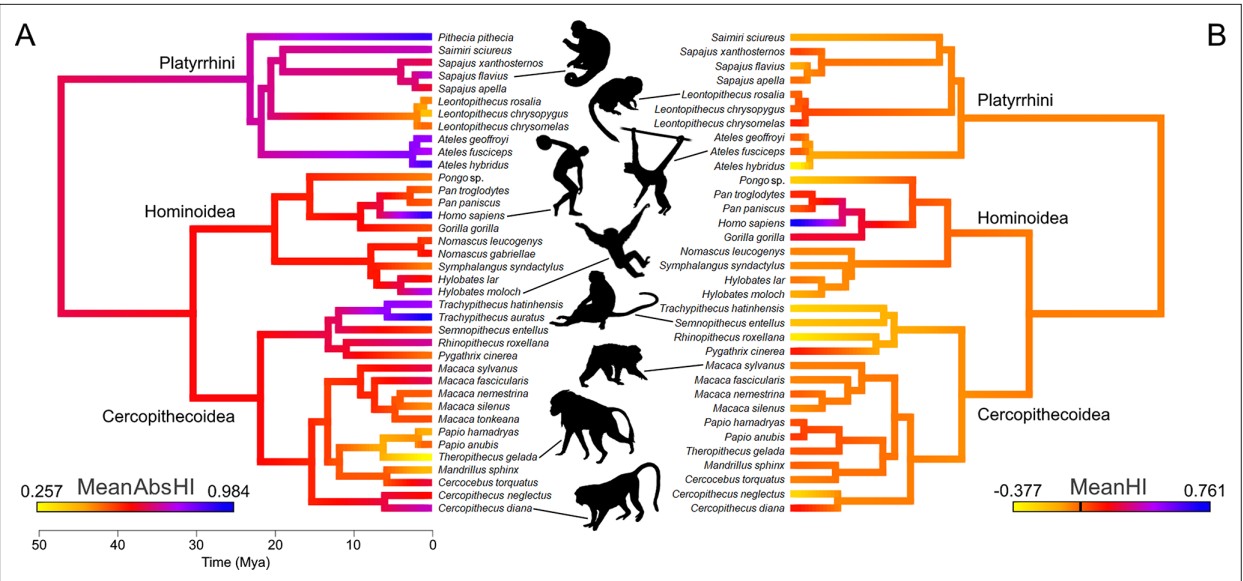

**Figure 1.** A color-coded phylogeny of hand preferences in anthropoid primates. The strength (**A**) and direction (**B**) of laterality, expressed by the mean absolute handedness index (MeanAbsHI) and the mean handedness index (MeanHI; 0 is marked by the black bar on the color scale), respectively, calculated for each species and inferred for each tree node by maximum likelihood estimates. Silhouettes by Kai R Caspar, except *Ateles* (by Yan Wong, public domain) and *Homo* (public domain).

*2008*; *Meguerditchian et al., 2015*; *Caspar et al., 2018*). Bimanual actions that involve both hands and which spontaneously occur during food manipulation or tool use are particularly suitable to detect hand preferences (*Meguerditchian et al., 2013*), while non-gestural unimanual actions such as grasping often do not elicit pronounced group- or individual-level lateralized responses (*Papademetriou et al., 2005*; *Rogers, 2009*; *Hopkins et al., 2013b*). However, habitual tool use is found in only a few lineages of primates (*Musgrave and Sanz, 2018*) and thus is only of limited use for comparative approaches. Therefore, bimanual actions related to foraging present themselves as suitable candidates for comparative studies on hand preferences in these animals.

A simple and widely applicable experiment to determine anthropoid primate hand preferences is the so-called tube task (*Hopkins, 1995*): A subject is handed a PVC tube filled with desired food. To extract it, one hand must hold the tube while the other has to engage in the more complicated action of retrieving the food mesh, thereby revealing biases in hand use dominance. Results from the tube task have been demonstrated to robustly correlate with hemispheric asymmetries in various primate groups (New World monkeys: *Phillips and Sherwood, 2005*; Old World monkeys: *Margiotoudi et al., 2019*; apes: *Hopkins and Cantalupo, 2004*; *Dadda et al., 2006*) and to be intraindividually consistent across re-tests, even if these were separated by several years (*Hopkins et al., 2001*). Furthermore, its simplicity allows uniform testing of a wide range of species in captive as well as natural settings (*Zhao et al., 2012*).

Here, we compiled a comprehensive multispecies tube task dataset to test pertaining hypotheses on the evolution of primate manual lateralization by means of phylogenetically informed modelling. This way we provide a broad comparative perspective on the origins of human right-handedness in the context of object manipulation.

## Results

Lateralization strength (MeanAbsHI) but not direction (MeanHI) displays a strong phylogenetic signal among anthropoids (*Figure 1*). Accordingly, MeanAbsHI ($\lambda$ =0.89, $p_{\text{likelihood-ratio test}}$ = 0.03) varied substantially between lineages but was often stable within groups of closely related taxa. MeanHI on the other hand showed random fluctuations around zero ($\lambda$ <0.001; $p_{\text{likelihood-ratio test}}$ = 1), and notable population level shifts toward either right- or left-hand preferences were uncommon (see below).

Species-level distributions of hand preferences are summarized in *Table 1* and are visualized at genus level in *Figure 2*.

Ancestral hand preference strength was modelled to have been similarly moderate in the stem lineages of hominoids (AbsHI = 0.606) and cercopithecoids (AbsHI = 0.627), while it was inferred to be higher in the ancestral platyrrhines (AbsHI = 0.740; *Figure 1*, *Supplementary file 1*). We found hand preference strength to be particularly weak among some species in the Papionina clade (baboons and their relatives) and to be least expressed in the gelada (*Theropithecus gelada* – MeanAbsHI = 0.257). The strongest preferences were found in humans (MeanAbsHI = 0.943), langurs of the genus *Trachypithecus* (MeanAbsHI = 0.868), and spider monkeys (*Ateles* spp. – MeanAbsHI = 0.831). Very pronounced individual preferences were also found in white-faced sakis (*Pithecia pithecia* – MeanAbsHI = 0.934) but our sample size for this species was notably small (n=7). In consequence, all aforementioned taxa included only very few, if any, ambipreferent individuals. Besides the saki genus *Pithecia*, the South Asian colobines *Rhinopithecus* and *Trachypithecus* were the only genera in which no ambipreferent individuals were found. Direction of manual lateralization was far more uniform across the anthropoid radiation than strength (*Figure 1*; *Supplementary file 1*). No species approached the extreme direction bias of humans (compare *Figure 2*), and only 2 of 37 non-human species exhibited a significant population-level bias as indicated by the one-sample t-test after correcting for multiple testing (4 of 37 when no correction was applied), namely gorillas and chimpanzees. Thus, such biases were restricted to the African ape lineage. After correcting for multiple testing, intraspecific frequencies of ambipreferent individuals, right-, and left-handers, as indicated by the chi-square goodness-of-fit test, differed significantly from the superordinate taxon estimate in four non-human species. These were chimpanzees, which are right-handed at the population level, and the predominately left-handed golden snub-nosed monkeys (*Rhinopithecus roxellana*), as well as geladas and black lion tamarins (*Leontopithecus chrysopygus*), both of which encompass a large proportion of ambipreferent individuals (*Table 1*). When omitting correction, respective biases were more frequent and found in 12 non-human species across the three major clades studied. Species- and genus-level results closely corresponded to each other (*Table 1*).

Our selected predictors for phylogenetic generalized least squares (PGLS) models showed a mixed and overall weak performance in explaining expression patterns of hand preference strength and direction in anthropoids (*Table 2*, *Figure 3*). For the initial model on lateralization direction, we found a significant effect of ecology, with terrestrial species displaying a right-hand bias compared to arboreal ones (p=0.04; *Table 2A*). However, when humans were removed from the model, this effect was merely recovered as a non-significant trend (p=0.07; *Table 2B*). Other predictors had no significant effect on lateralization direction, regardless of whether humans were considered in the model or not (p>0.2, *Table 2A and B*). When humans were included (*Table 3A*), models encompassing the component ecology as well as some considering brain size outperformed the lateralization direction null model. When humans were omitted from the analysis, the null model was solely outperformed by one that exclusively included the ecological component, and only slightly so (ΔAICc = 1.23; *Table 3B*), indicating a notable bias derived from the extreme right-handedness of our species. Thus, whereas habitual tool use and absolute brain size clearly do not influence the direction of lateralization among anthropoids in general, the analyses provide evidence for a weak but detectable effect of ecology in non-human taxa. Such ecological signatures were found to be of somewhat greater relevance for patterns of lateralization strength. Here, a significant negative effect of a terrestrial lifestyle was found (p=0.04; see *Table 2C*). In line with that, models including the component ecology consistently outperformed the null model, which was not the case for those including only tool use and/or brain size (*Table 3C*). Still, even the accuracy of the model relying on ecology alone exceeded that of the null model only moderately (ΔAICc = 2.53; *Table 3C*). Thus, terrestrial anthropoids tend to show weaker hand preferences than arboreal ones while there is no correlation with brain size or habitual tool use for this trait.

At the individual level, Bayesian models showed that neither age nor sex had an influence on lateralization direction when the total sample was concerned (the respective credible intervals overlapped with zero; *Supplementary file 2*). However, we recovered a notable effect of age on lateralization direction in the hominoid subsample exclusively, with subadults tending more toward left-handedness than adults (credible interval = –0.28 to –0.01; *Supplementary file 2*). We observed a different pattern for lateralization strength. Here, an effect of age but again not sex was recovered for both the total

**Table 1.** Hand preferences of anthropoid species as recovered by the tube task. Bold numbers indicate significant results. Results marked with an asterisk (*) remain significant after Bonferroni correction.

| Species | N | # Left (%) | # Right (%) | # Ambipreferent (%) | MeanHI | MeanAbsHI | Species direction bias (HI), p value | Species L/R/A distribution, p value | nGenus | Genus direction bias (HI), p value | Genus L/R/A distribution, p value |
|---|---|---|---|---|---|---|---|---|---|---|---|
| Ateles fusciceps | 46 | 20 (43.5) | 22 (47.8) | 4 (8.7) | 0.063 | 0.798 | 0.618 | 0.288 | | | |
| Ateles geoffroyi | 23 | 10 (43.5) | 11 (47.8) | 2 (8.7) | 0.061 | 0.829 | 0.748 | 0.536 | | | |
| Ateles hybridus | 18 | 13 (72.2) | 5 (27.8) | 0 | -0.377 | 0.917 | 0.086 | **0.018** | 87 | 0.759 | **0.031** |
| Cercocebus torquatus | 31 | 13 (41.9) | 11 (35.5) | 7 (22.6) | -0.029 | 0.665 | 0.832 | 0.836 | 31 | 0.832 | 0.836 |
| Cercopithecus diana | 20 | 7 (35) | 10 (50) | 3 (15) | 0.178 | 0.755 | 0.339 | 0.836 | | | |
| Cercopithecus neglectus | 25 | 14 (56) | 7 (28) | 4 (16) | -0.258 | 0.621 | 0.061 | 0.140 | 45 | 0.572 | 0.222 |
| Gorilla gorilla | 76 | 17 (22.4) | 41 (53.9) | 18 (23.7) | 0.248 | 0.541 | **<0.001*** | **0.007** | 76 | **<0.001*** | **0.007** |
| Homo sapiens | 127 | 12 (9.5) | 111 (87.4) | 4 (3.1) | 0.761 | 0.943 | **<0.001*** | **<0.001*** | 127 | **<0.001*** | **<0.001*** |
| Hylobates lar | 36 | 17 (47.2) | 16 (44.5) | 3 (8.3) | -0.011 | 0.614 | 0.924 | 0.182 | | | |
| Hylobates moloch | 22 | 11 (50) | 8 (36.4) | 3 (13.6) | -0.115 | 0.799 | 0.540 | 0.552 | 58 | 0.612 | 0.125 |
| Leontopithecus chrysomelas | 30 | 7 (23.3) | 12 (40) | 11 (36.7) | 0.151 | 0.514 | 0.171 | **0.012** | | | |
| Leontopithecus chrysopygus | 15 | 3 (20) | 4 (26.7) | 8 (53.3) | 0.039 | 0.350 | 0.744 | **0.001*** | | | |
| Leontopithecus rosalia | 28 | 10 (35.7) | 8 (28.6) | 10 (35.7) | 0.022 | 0.502 | 0.850 | **0.033** | 73 | 0.241 | **<0.001*** |
| Macaca fascicularis | 20 | 8 (45) | 10 (45) | 2 (10) | -0.036 | 0.684 | 0.835 | 0.233 | | | |
| Macaca nemestrina | 29 | 9 (31) | 11 (37.9) | 9 (31) | 0.035 | 0.527 | 0.768 | 0.750 | | | |
| Macaca silenus | 35 | 14 (40) | 9 (25.7) | 12 (34.3) | -0.051 | 0.467 | 0.596 | 0.328 | | | |
| Macaca sylvanus | 24 | 12 (50) | 10 (41.7) | 2 (8.3) | -0.025 | 0.670 | 0.873 | 0.129 | | | |
| Macaca tonkeana | 14 | 5 (35.7) | 3 (21.4) | 6 (42.9) | -0.057 | 0.543 | 0.753 | 0.291 | 102 | 0.692 | 0.863 |
| Mandrillus sphinx | 32 | 6 (18.8) | 10 (31.2) | 16 (50) | 0.034 | 0.389 | 0.701 | **0.006** | 32 | 0.701 | **0.006** |
| Nomascus gabriellae | 10 | 5 (50) | 2 (20) | 3 (30) | -0.173 | 0.618 | 0.465 | 0.436 | | | |
| Nomascus leucogenys | 26 | 9 (34.6) | 11 (42.3) | 6 (23.1) | -0.031 | 0.555 | 0.818 | 0.869 | 36 | 0.539 | 0.805 |
| Pan paniscus | 118 | 50 (42.4) | 51 (43.2) | 17 (14.4) | 0.044 | 0.529 | 0.431 | 0.237 | | | |
| Pan troglodytes | 536 | 155 (28.9) | 266 (49.6) | 115 (21.5) | 0.133 | 0.507 | **<0.001*** | **<0.001*** | 654 | **<0.001*** | **<0.001*** |

*Table 1 continued on next page*

Table 1 continued

| Species | N | # Left (%) | # Right (%) | # Ambipreferent (%) | MeanHI | MeanAbsHI | Species direction bias (HI), p value | Species L/R/A distribution, p value | nGenus | Genus direction bias (HI), p value | Genus L/R/A distribution, p value |
|---|---|---|---|---|---|---|---|---|---|---|---|
| Papio anubis | 84 | 27 (32.1) | 41 (48.8) | 16 (19.1) | 0.108 | 0.527 | 0.102 | 0.073 | | | |
| Papio hamadryas | 24 | 6 (25) | 7 (29.2) | 11 (45.8) | 0.066 | 0.408 | 0.533 | 0.082 | 108 | 0.079 | 0.239 |
| Pithecia pithecia | 7 | 5 (71.4) | 2 (28.6) | 0 | −0.385 | 0.934 | 0.312 | 0.221 | 7 | NA | 0.221 |
| Pongo sp. | 47 | 27 (57.5) | 9 (19.1) | 11 (23.4) | −0.225 | 0.487 | **0.006** | **0.012** | 47 | **0.006** | **0.012** |
| Pygathrix cinerea | 18 | 6 (33.3) | 10 (55.6) | 2 (11.1) | 0.165 | 0.499 | 0.268 | 0.196 | 18 | 0.268 | 0.196 |
| Rhinopithecus roxellana | 24 | 17 (70.8) | 7 (29.2) | 0 | −0.319 | 0.729 | **0.040** | **<0.001*** | 24 | **0.040** | **<0.001*** |
| Saimiri sciureus | 36 | 21 (58.4) | 14 (38.9) | 1 (2.7) | −0.119 | 0.757 | 0.382 | **0.031** | 36 | 0.382 | **0.031** |
| Sapajus apella | 25 | 11 (44) | 10 (40) | 4 (16) | −0.028 | 0.687 | 0.854 | 0.961 | | | |
| Sapajus flavius | 21 | 10 (47.6) | 7 (33.3) | 4 (19) | −0.130 | 0.769 | 0.495 | 0.755 | | | |
| Sapajus xanthosternos | 34 | 14 (41.2) | 15 (44.1) | 5 (14.7) | 0.089 | 0.677 | 0.492 | 0.906 | 80 | 0.922 | 0.905 |
| Semnopithecus entellus | 30 | 15 (50) | 7 (23.4) | 8 (26.6) | −0.184 | 0.560 | 0.110 | 0.315 | 30 | 0.110 | 0.315 |
| Symphalangus syndactylus | 31 | 11 (35.5) | 9 (29) | 11 (35.5) | −0.048 | 0.482 | 0.663 | 0.118 | 31 | 0.663 | 0.118 |
| Theropithecus gelada | 38 | 6 (15.8) | 6 (15.8) | 26 (68.4) | 0.053 | 0.257 | 0.326 | **<0.001*** | 38 | 0.326 | **<0.001*** |
| Trachypithecus auratus | 8 | 5 (62.5) | 3 (37.5) | 0 | −0.256 | 0.984 | 0.499 | 0.176 | | | |
| Trachypithecus hatinhensis | 18 | 11 (61.1) | 7 (38.9) | 0 | −0.248 | 0.817 | 0.219 | **0.023** | 26 | 0.153 | 0.004 |

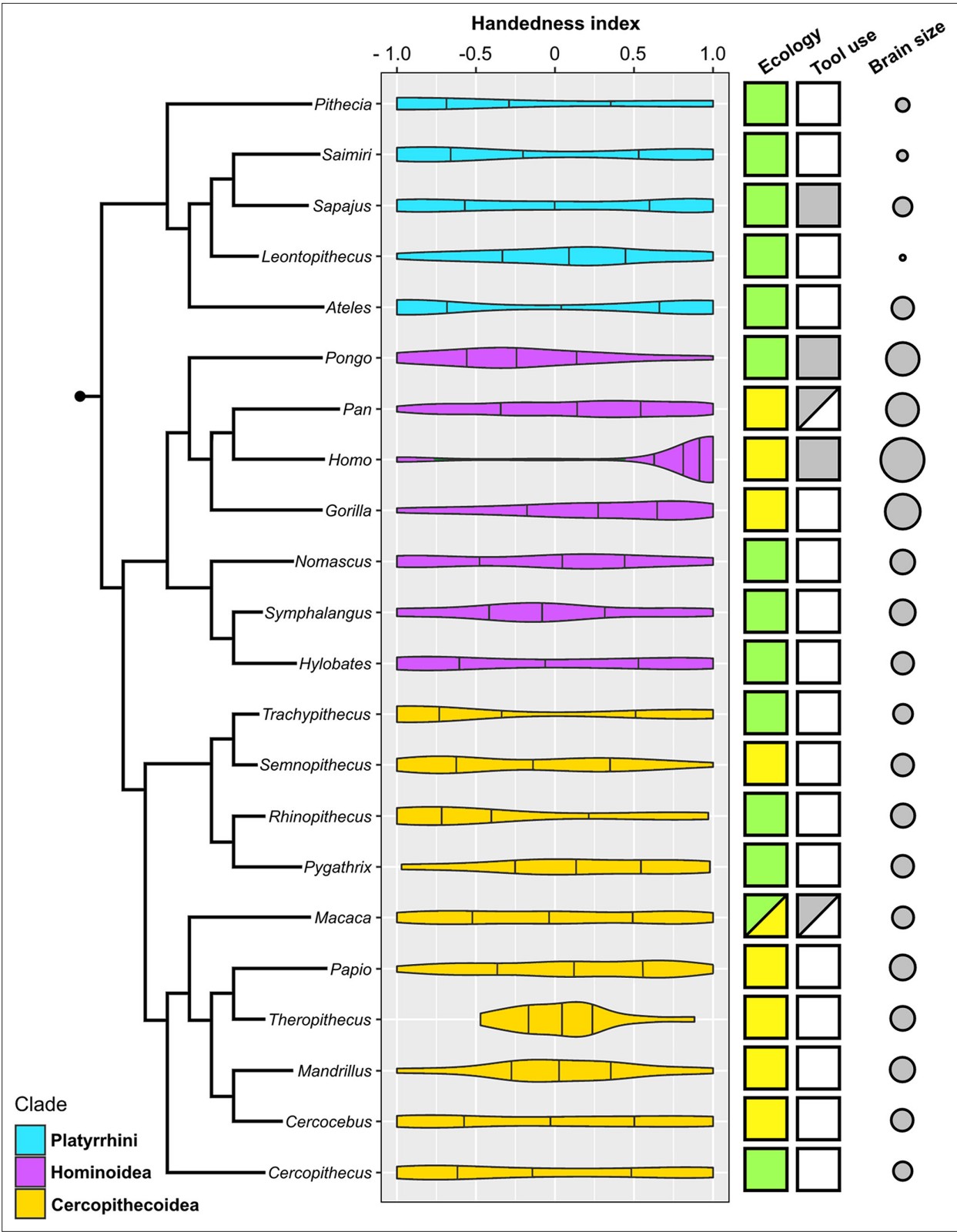

**Figure 2.** Violin plots of hand preference distribution in 22 genera of anthropoid primates and the genus-specific expression of three potential biological correlates (ecology, habitual foraging-related tool use, and absolute brain size). Attributions only apply to the species that represent the respective genus within our sample. Color coding: Ecology – green: arboreal, yellow: terrestrial; Habitual tool use – gray: present; white: absent. Brain size is visualized here as the log-transformed genus average of female endocranial volume.

**Table 2.** Conditional average of phylogenetic generalized least squares (PGLS) model coefficients for lateralization strength and direction in anthropoid primate species.
Bold numbers indicate significant results. VIF = variable inflation factor.

A: **Conditional PGLS model average for lateralization direction, including humans**

| Predictor | Estimate | Std. error | VIF | p value |
|---|---|---|---|---|
| Ecology (terrestrial lifestyle) | 0.153 | 0.072 | 1.499 | 0.040 |
| Tool use (present) | 0.104 | 0.082 | 1.164 | 0.220 |
| $Log_{10}$ brain size | 0.050 | 0.043 | 1.612 | 0.254 |

B: Conditional PGLS model average for lateralization direction, excluding humans

| Predictor | Estimate | Std. error | VIF | p value |
|---|---|---|---|---|
| Ecology (terrestrial lifestyle) | 0.108 | 0.056 | 1.419 | 0.067 |
| $Log_{10}$ brain size | −0.020 | 0.037 | 1.405 | 0.601 |
| Tool use (present) | 0.003 | 0.068 | 1.098 | 0.962 |

C: Conditional PGLS model average for lateralization strength

| Predictor | Estimate | Std. error | VIF | p value |
|---|---|---|---|---|
| Ecology (terrestrial lifestyle) | −0.143 | 0.067 | 1.813 | **0.040** |
| Tool use (present) | 0.060 | 0.070 | 1.235 | 0.402 |
| $Log_{10}$ brain size | 0.035 | 0.047 | 1.997 | 0.468 |

sample (credible interval = −0.11 to −0.05) and the hominoid subsample (credible interval = −0.16 to −0.02), with weaker lateralization in subadults compared to adults. Such an influence of age was not detectable in neither platyrrhines nor cercopithecoids when these taxa were considered separately (credible intervals overlapping with zero; *Supplementary file 2*).

## Discussion
### General

Our study provides the first quantitative phylogenetic perspectives on hand preferences in monkeys, apes, and humans. While population-level lateralization strength is highly varied among anthropoid primates and often distinctly expressed in specific lineages, direction fluctuates irrespective of phylogeny and appears comparatively uniform. Indeed, significant population-level biases in the latter are notably rare, both at the species and genus level. After expanding the sample size for

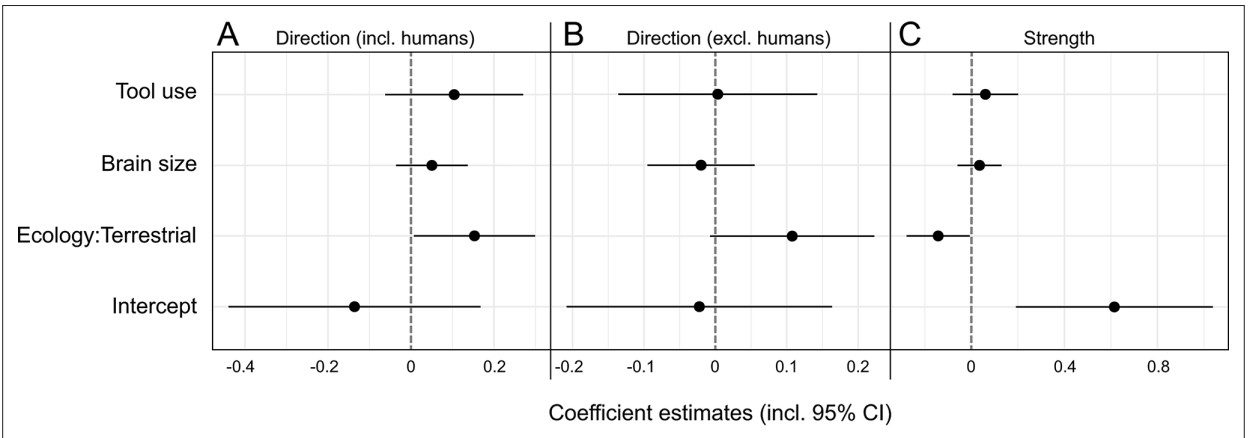

**Figure 3.** Visualization of phylogenetic generalized least squares (PGLS) coefficient estimates (including 95% confidence intervals) for the influence of brain size, tool use, and ecology on lateralization direction (**A, B**) as well as strength (**C**) in anthropoid primates. Two models for lateralization direction were computed, one including (**A**), the other one excluding humans (**B**). The strength model encompassed humans as well (**C**).

**Table 3.** Results of phylogenetic generalized least squares (PGLS) model averaging for lateralization direction (considering the inclusion and exclusion of humans) and strength.

Null models are shown in italics. Df. = degrees of freedom. AICc = second-order Akaike information criterion.

A: **PGLS model for lateralization direction, including humans**

| Components | Df. | AICc | ΔAICc | Weight |
|---|---|---|---|---|
| Ecology | 3 | –13.60 | 0 | 0.28 |
| Ecology, tool use | 4 | –13.37 | 0.23 | 0.25 |
| Brain size | 3 | –12.22 | 1.38 | 0.14 |
| Ecology, brain size | 4 | –12.07 | 1.54 | 0.13 |
| Ecology, tool use, brain size | 5 | –10.87 | 2.73 | 0.07 |
| *(NULL)* | *2* | *–10.37* | *3.23* | *0.05* |
| Tool use, brain size | 4 | –10.20 | 3.41 | 0.05 |
| Tool use | 3 | –9.48 | 4.12 | 0.04 |

B: PGLS model for lateralization direction, excluding humans

| Components | Df. | AICc | Δ | Weight |
|---|---|---|---|---|
| Ecology | 3 | –30.60 | 0 | 0.37 |
| *(NULL)* | *2* | *–29.37* | *1.23* | *0.20* |
| Ecology, brain size | 4 | –28.82 | 1.78 | 0.15 |
| Ecology, tool use | 4 | –28.01 | 2.59 | 0.10 |
| Tool use | 3 | –26.98 | 3.62 | 0.06 |
| Brain size | 3 | –26.97 | 3.63 | 0.06 |
| Ecology, tool use, brain size | 5 | –26.16 | 4.44 | 0.04 |
| Tool use, brain size | 4 | –24.42 | 6.18 | 0.02 |

C: PGLS model for lateralization strength

| Components | Df. | AICc | ΔAICc | Weight |
|---|---|---|---|---|
| Ecology | 3 | –8.84 | 0 | 0.35 |
| Ecology, brain size | 4 | –7.85 | 0.99 | 0.21 |
| Ecology, tool use | 4 | –7.33 | 1.51 | 0.16 |
| *(NULL)* | *2* | *–6.31* | *2.53* | *0.10* |
| Ecology, brain size, tool use | 5 | –5.46 | 3.38 | 0.06 |
| Tool use | 3 | –5.05 | 3.80 | 0.05 |
| Brain size | 3 | –4.14 | 4.71 | 0.03 |
| Tool use, brain size | 4 | –3.19 | 5.66 | 0.02 |

some species in which such biases have been previously reported based on the tube task (siamang – *Morino et al., 2017*; de Brazza's monkey – *Schweitzer et al., 2007*), we failed to replicate significant deviations from a chance distribution (even if not correcting for multiple testing). The only taxon in which significant hand use biases for bimanual manipulation occur frequently is constituted by the great apes and humans. Nevertheless, since sample sizes for species in this group are consistently and conspicuously large, statistical analyses performed on them (in particular the commonly applied one-sample t-test) will have higher power compared to tests done on taxa represented by a smaller number of individuals. It is therefore premature to assume that hominids display qualitatively different population-level lateralization patterns than other primates based on this statistical parameter. In fact,

we would like to stress that due to this issue, simply determining the presence or absence of significant population-level handedness is of little merit for comparative studies, since species with vastly different hand preference distributions might fall into either category dependent on the available sample sizes (*Figure 2*). Looking at the population-level frequencies of right-handed, left-handed, and ambipreferent individuals, non-hominoid species such as the golden snub-nosed monkey (70.8% left-handers, 0% ambipreferents, MeanHI: –0.319) and the brown spider monkey (72.2% left-handers, 0% ambipreferents, MeanHI: –0.377) approach a human-like skew more than any great ape species does, albeit in the contrary direction to lateralization in our species (approximated herein as encompassing 87.3% right-handers, 3.1% ambipreferents, MeanHI: 0.761). Whether the hand preference patterns recovered for these rather poorly sampled monkeys are indeed representative for the respective species, needs to be clarified by future studies encompassing greater numbers of individuals.

Our finding that hand preference strength is generally weaker in juveniles compared to adults replicates results from several studies relying on smaller sample sizes (e.g., *Westergaard and Suomi, 1996*; *Zhao et al., 2012*). The result that subadult non-human hominoids, but not other anthropoids, are notably stronger left-handed than adults, likely derives from the aforementioned fact that juveniles are more weakly lateralized overall in conjunction with the composition of our Bayesian model sample. Therein, Western gorillas and bonobos (*Pan paniscus*), both of which tend to be right-handed as adults (*Hopkins et al., 2011*), are well represented across age groups. Again, we suggest that an expanded dataset encompassing both an increased number of individuals and species could potentially level out the lateralization differences between (great) apes and other anthropoids observed in our study.

The fact that population-level hand preference fluctuates without phylogenetic and with rather weak ecological signatures among anthropoids suggests that there are no strong directional selective pressures acting on this trait, different from what pertaining hypotheses predict (see below). On the other hand, population-level lateralization strength is more variable but consistent among closely related taxa, thus exhibiting a strong phylogenetic signal. PGLS modelling demonstrated a significant negative effect of a terrestrial lifestyle on hand preference strength, indicating a relevant but previously undescribed influence of ecology. It appears intuitive that terrestrial taxa tend to be less lateralized than arboreal ones, since the latter often need to flexibly stabilize their body in the canopy, for instance while foraging. Accordingly, one hand will be preferably used to provide such support, but whether the left or the right one adopts this function seems to be arbitrary. The fact that these lateralization tendencies were found in zoo-housed primates that often adopt locomotor regimes very different from their wild conspecifics (e.g., captive spider monkeys and gibbons spend considerable amounts of time moving and feeding on the ground) suggests a significant innate component to these patterns. However, within ecologically uniform groups the variability of hand preference strength can still be notable, at times even among closely related taxa (compare, e.g., Javan gibbon and siamang), pointing at further important but yet unidentified biological influences being at play as well. Accordingly, the explanatory power of PGLS models for lateralization strength that considered the component ecology, only moderately exceeded that of a null model (compare *Table 3C*). Nevertheless, given the great variability of lateralization strength among anthropoids and its ties to phylogeny, this aspect of manual lateralization should receive more research attention in the future. In the past, most work and evolutionary considerations regarding primate handedness have instead focused on lateralization direction, surely for anthropocentric reasons. As we attempt to show here, however, the explanatory power of these in parts very long-lived conjectures appears to be remarkably limited.

## Testing prevalent hypotheses

Our data does not unambiguously support any of the tested hypotheses on hand preference evolution in primates. The traditional POH assumes right-hand tendencies for manipulation across anthropoid taxa. However, we found that anthropoid population-level lateralization is in most cases not notably shifted into either direction, with a slight majority of species displaying a weak left-hand bias (21 of 38 species). It is important to note that the recovered correlation between arboreality and hand preference strength does not corroborate any version of the POH, as they focus exclusively on lateralization direction. The novel POH assumes that terrestrial non-human primates tend to be right-handed, while arboreal ones tend to be left-handed, a prediction that gains weak support from our data. But while the mean handedness indices (HIs) of terrestrial and arboreal non-human species appear dissimilar, they do not deviate strongly from zero and there is a notable overlap in variation between the two

ecological groups (MeanHI$_{arboreal}$: –0.08, SD: 0.16; MeanHI$_{terrestrial}$: 0.04, SD: 0.10). Thus, the predictive power of the hypothesis is markedly low, which is reflected by our modelling results (*Tables 2 and 3*). Admittedly though, improved sampling both within and across species might consolidate ecological patterns in the future, so that we certainly do not want to dismiss relevant effects of lifestyle on lateralization direction at this point.

However, even if the predictions of the novel POH should become confirmed eventually, this would not necessarily validate its theoretical framework. First, the hypothesis does not provide an evolutionary mechanism for why population-level hand preference patterns should be coupled with positional behavior. This represents a considerable conceptual shortcoming that needs to be addressed in the future. Second, it is important to note that the evolutionary scenario proposed by both versions of the POH is outdated and therefore should not be perpetuated without explicitly stating its flaws. According to the POH, small-bodied bushbabies (genus *Galago*) are suitable models for early primates, since they would be 'the most direct descendants of the earliest forms' (*MacNeilage, 2007*). Because contemporary studies had suggested a left-hand bias for prey grasping in bushbabies, such a pattern was also assumed for primate ancestors in the paper that introduced the original POH (*MacNeilage et al., 1987*). However, these assumptions have always been speculative and become problematic in light of more recent data. Importantly, there is no convincing evidence for preferably left-handed grasping in the genus *Galago,* or other galagids, anymore (*Papademetriou et al., 2005*). Furthermore, bushbabies represent a remarkably derived radiation of strepsirrhines (*Rasmussen and Nekaris, 1998*) and are thus no suitable ecological models for the common ancestor of modern primates. Current evidence suggests that both the earliest primates and the ancestors of the anthropoid clade were omnivorous arboreal quadrupeds with moderate leaping ability (*Silcox et al., 2009*; *Gebo, 2011*; *Sussman et al., 2013*) and possibly diurnal habits (*Tan et al., 2005*; *Ankel-Simons and Rasmussen, 2008*). Thus, they were extremely different from extant galagos. Our results indicate that the common ancestor of anthropoids did not display notable population-level hand preferences for manipulation, nor that such biases are common among extant monkeys and apes. Their potential occurrence among strepsirrhines still needs to be comparatively assessed, though. While lemurs (different from galagos) indeed appear to show a consistent left-hand preference for unimanual reaching, this pattern is not recovered in bimanual tasks (*Papademetriou et al., 2005*; *Regaiolli et al., 2016*; *Batist and Mayhew, 2020*), contradicting the POH (*MacNeilage, 2007*). In conclusion, the evolutionary scenario proposed by both versions of the POH is unsupported by current data. Moreover, our findings contradict the predictions of the original POH and only provide a very fragile empirical basis for the novel version of the hypothesis. This leads us to challenge the status of the POH as a keystone idea in the discourse on the evolution of primate manual lateralization.

We also found no effects of foraging-related tool use on neither direction nor strength of lateralization although our sample represented all primate lineages that include habitual tool users (*Musgrave and Sanz, 2018*). Surprisingly, we also did not recover notable influences of absolute brain size on hand preferences. An effect on lateralization strength was expected both on theoretical considerations (*Ringo et al., 1994*) and empirical evidence from studies investigating intra- and interspecific covariation of brain size and overall cortical lateralization (*Kong et al., 2018*; *Ardesch et al., 2021*). Why do anthropoid hand preferences not conform to these predictions? We cannot provide a satisfying answer to this question. It is possible that the effects of increased overall brain lateralization on hand preference expression turn out to be unexpectedly weak and are masked by yet unidentified neurological factors.

All in all, none of the hypotheses on primate handedness that we addressed were clearly corroborated by our results. Nevertheless, when discussing conflicts between prevalent ideas and our results, we also need to address general limitations of our study framework. For instance, we equate general hand preferences with tube task results. Although the tube task represents one of the best available behavioral assays for brain lateralization (*Dadda et al., 2006*; *Margiotoudi et al., 2019*), it is obvious that other hand use situations, for example communicative gesturing, need to be considered to arrive at a holistic understanding of primate handedness evolution (*Hopkins et al., 2013a*; *Becker et al., 2022*). Such an approach could also test how variable hand use consistency across contexts is among primates and whether ecological variables have an influence here as well. At the moment, it appears as if there is comparatively little consistency in manual preferences across different hand use situations in non-human primates (*Laska, 1996*; *Lilak and Phillips, 2008*; *Marchant and McGrew, 2013*; *Caspar*

*et al., 2018*). Still, most studies so far compare tasks of varying complexity in a few model species and cases of consistent hand use ('true handedness') have indeed been reported (*Diamond and McGrew, 1994*; *Hopkins et al., 2013a*; but note neuroanatomical evidence for a dissociation of lateralization in non-communicative actions and gestures in baboons – *Becker et al., 2022*).

Another limitation is posed by our sample composition. Both the number of species and subjects per species need to be increased to further validate the patterns communicated here. In particular, extended sampling of the speciose New World monkey families Pitheciidae and Callitrichidae would be desirable, to make the inferences presented herein more robust. To allow for proper predictions on hand preferences in anthropoid ancestors and early crown-group primates, additional tests with tarsiers and strepsirrhines would be crucial. Our experience suggests that at least pitheciids and lemurs only reluctantly engage in the tube task so that it might be advisable to apply different bimanual testing schemes in these groups. In lemurs, puzzle boxes have been employed as such: The animals open the lid of a box with one hand while the other one is retrieving food stored within (*Regaiolli et al., 2016*; *Batist and Mayhew, 2020*). Future studies need to check the functional equivalence of this approach with the tube task (which is not a trivial question, compare, e.g., *Lilak and Phillips, 2008*) to establish a set of behavioral assays that could be employed to study hand preferences in the whole primate order. These methods might then also be applied to other dexterous and ecologically variable mammalian groups, such as musteloid carnivorans (*Kitchener et al., 2017*), to test hypotheses on the evolution of manual laterality across a wider phylogenetic margin. Finally, we need to acknowledge the limitations of our phylogenetic modelling approach. In particular, the binary coding of *ecology* (arboreal vs. terrestrial) obviously simplifies the remarkable spectrum of positional behaviors found among anthropoids. Future studies might explore alternative strategies to statistically code this multifaceted variable.

## The evolutionary issue of human handedness

In line with previous research, we found human right-handedness to be unparalleled among primates. We want to stress, however, that humans only deviate markedly from all other taxa in direction and not in strength of lateralization for bimanual manipulation. When it comes to the latter, the human condition is at least approached by groups such as leaf monkeys and spider monkeys. Perhaps surprisingly, handedness strength in great apes is modest in comparison (*Table 1*). Still, humans are highly deviant among predominately terrestrial primates in displaying such strong individual hand preferences. Whether this difference relates to bipedal locomotion, which has often been championed as a correlate of human handedness (*Westergaard et al., 1998*; *Cashmore et al., 2008*; *Prieur et al., 2019*), is open for debate. Since no other extant primate shows similar adaptations to terrestrial bipedalism, the validity of this assumption is hard to test in the framework of comparative approaches (but see *Giljov et al., 2015*). Interestingly, quadrupedal primates tend to exhibit stronger hand preferences when adopting the relatively unstable bipedal posture (*Westergaard et al., 1998*). Still, whether this finding has any evolutionary implications remains unclear and it should be emphasized that although humans are bipeds, a high percentage of complex manual actions, including numerous examples of bimanual manipulation and tool use, are not (and never have been) habitually performed in a bipedal posture. In any case, our results suggest that bipedalism is at least not a prerequisite to evolve strong hand preferences in anthropoid primates.

When turning to lateralization direction, however, the statement of *Corballis, 1987*, remains valid: some non-primate vertebrates approach humans more closely in population-level handedness than their simian relatives do. Apart from humans, extreme forms of vertebrate limb use biases are known from parrots (*Kaplan and Rogers, 2021*) and ground-living kangaroos (*Giljov et al., 2015*). Why these very different groups converge in this respect remains enigmatic. So why do humans stand out among the primate order when it comes to handedness direction? The limited insights gained by comparative behavioral studies, including this one, may suggest that the extreme right-handedness of humans is a trait that evolved due to unique neurophysiological demands not experienced by other primates. *Frost, 1980*, already pointed out that humans' outstanding proficiency in tool use and manufacture should be considered a significant influence on handedness evolution. Thus, not foraging-related tool use per se, but the unique way in which it became immersed into complex human behaviors might have influenced overall brain lateralization in our lineage. In line with that, areas of the prefrontal cortex involved in motor cognition are structurally derived in humans and differ significantly from their

homologs in apes and monkeys (*Hecht et al., 2015b*; *Barrett et al., 2020*). However, specializations of both the right and the left hemisphere are determining human-specific tool use proficiency and motor planning, apparently with particular involvement of the right inferior frontal gyrus (*Ramayya et al., 2010*; *Hecht et al., 2015b*; *Hecht et al., 2015a*). Postulating that hominin tool use and right-handedness evolved in tandem is therefore not straight-forward.

Besides that, there is of course the notion of coevolution between language and handedness, which might explain human-specific patterns of population-level manual lateralization. For this hypothesis to be convincing, the development and function of neural substrates controlling vocal behavior and those regulating manual motor control would need to be uniquely intertwined in humans. Indeed, the connectivity of the arcuate fasciculus, a tract critically involved in language processing and production, is highly derived in humans, suggesting important qualitative differences to other species (*Rilling et al., 2008*; but see *Barrett et al., 2020*, for other elements relevant for language production which are conserved across catarrhine primates). Nevertheless, how such neuroanatomical traits could functionally relate to population-level handedness remains totally unclear. In fact, despite the popularity of the idea, a link between handedness and language processing that goes beyond superficial left-hemisphere collateralization in right-handers (not even in the majority of left-handers) is far from evident (*Fitch and Braccini, 2013*). To defend an evolutionary connection between these phenomena, pleiotropic or otherwise functionally linked genes influencing the development of both language areas and those related to handedness would need to be identified. So far, this has not been accomplished and current evidence suggests that language and handedness are largely independent on various structural levels (*Ocklenburg et al., 2014*; *Schmitz et al., 2017b*). Hence, despite the continuing efforts to unravel the evolution of human right-handedness, including the ones made by us herein, it remains an essentially unsolved issue of human cognitive evolution.

## Conclusions

We recovered highly variable patterns of hand preference strength in anthropoid primates, which correlate with ecology and phylogeny. In contrast to this, no phylogenetic signal and weaker ecological effects were found for lateralization direction, and few species exhibit significant population-level hand preferences. We tested three pertaining conjectures on primate handedness evolution, the POH, tool use, and brain size hypotheses, but none were unambiguously corroborated by our data. Hypotheses on the evolution of primate hand preferences should put a stronger focus on manual lateralization strength rather than direction to address the phylogenetic patterns described herein. However, additional datasets on primates and potentially non-primate mammals are needed to robustly inform novel concepts. By relying on standardized testing paradigms, such as the tube task, researchers can effectively build on our as well as others' results and expand multispecies datasets for further comparative phylogenetic studies. Although we are convinced that such approaches could significantly improve our understanding of general trends in the evolution of primate hand preferences, the unusual autapomorphic handedness pattern of humans will very likely require explanations that cannot be derived from such comparative behavioral studies. The evolutionary underpinnings of handedness expression in our species remain enigmatic.

## Materials and methods
### Subjects

We analyzed the expression of hand preferences for object manipulation in the tube task, as well as potential factors influencing their evolution, for a dataset of anthropoid primates (infraorder Anthropoidea: New World monkeys [Platyrrhini], Old World monkeys [Cercopithecoidea], and apes [Hominoidea]) from 38 species. Data from 501 individuals belonging to 26 primate species were collected in the tube task paradigm (see below) between September 2017 and May 2020 in 39 institutions in Europe, Brazil, and Indonesia (*Table 4*). Of these species, 14 had never been tested in the tube task before. Additional datasets were drawn from the literature, resulting in a total sample of 1786 individuals from 38 species and 22 genera, covering all anthropoid primate families except Aotidae. Data for humans were derived from *Cochet and Vauclair, 2012*. In this study, participants had to use one hand to repeatedly retrieve pieces of paper out of a plastic cylinder while the other one had to tilt and stabilize the receptacle. We considered this bimanual testing paradigm as functionally equivalent to

**Table 4.** Composition of the study sample, listing taxonomic identity, sex, age, and origin of subjects. See cited studies for locations of individuals drawn from the literature.

| Family | Species | # Subjects tested | # Subjects drawn from literature* | Total sample | # Adult females | # Adult males | # Subadult females | # Subadult males | # Unsexed subadults | Locations for subjects in this study |
|---|---|---|---|---|---|---|---|---|---|---|
| Atelidae | Ateles fusciceps | 37 | 9§ | 46 | 30 | 11 | 3 | 2 | 0 | Berlin (Zoo), Doué-la-Fontaine, Landau, Mulhouse, Munich, Osnabrück, Wuppertal |
| Atelidae | Ateles geoffroyi | 9 | 14¶ | 23 | 12 | 9 | 0 | 2 | 0 | Basel, Karlsruhe |
| Atelidae | Ateles hybridus | 18 | | 18 | 10 | 7 | 0 | 1 | 0 | Doué-la-Fontaine, Erfurt, Frankfurt, Neuwied, Stuttgart |
| Callitrichidae | Leontopithecus chrysomelas | 30 | | 30 | 11 | 16 | 2 | 1 | 0 | Apeldoorn, Karlsruhe, Magdeburg, Mulhouse, São Paulo, Stuttgart, Wuppertal |
| Callitrichidae | Leontopithecus chrysopygus | 15 | | 15 | 6 | 9 | 0 | 0 | 0 | São Paulo |
| Callitrichidae | Leontopithecus rosalia | 28 | | 28 | 7 | 16 | 0 | 5 | 0 | Apeldoorn, Basel, Doué-la-Fontaine, Duisburg, Frankfurt, Heidelberg, Landau, Magdeburg, Münster, São Paulo |
| Cebidae | Saimiri sciureus | | 36** | 36 | 14 | 16 | 5 | 1 | 0 | |
| Cebidae | Sapajus apella | | 25††‡‡ | 25 | 10 | 11 | 0 | 4 | 0 | |
| Cebidae | Sapajus flavius | 3 | 18‡ | 21 | 7 | 9 | 2 | 3 | 0 | São Paulo |
| Cebidae | Sapajus xanthosternos | 16 | 18‡ | 34 | 11 | 19 | 1 | 2 | 1 | Apeldoorn, Magdeburg, Münster, Overloon |
| Cercopithecidae | Cercocebus torquatus | 18 | 13§ | 31 | 15 | 13 | 1 | 2 | 0 | Apeldoorn, Berlin (Tierpark), Karlsruhe, Münster |
| Cercopithecidae | Cercopithecus diana/roloway | 20 | | 20 | 9 | 7 | 3 | 1 | 0 | Amsterdam, Berlin (Tierpark), Doué-la-Fontaine, Duisburg, Heidelberg, Liberec, Mulhouse, Osnabrück |
| Cercopithecidae | Cercopithecus neglectus | 12 | 13§ §,¶¶ | 25 | 8 | 12 | 1 | 4 | 0 | Bekesbourne, Duisburg, Hannover, Overloon |
| Cercopithecidae | Macaca fascicularis | 12 | 8*** | 20 | 13 | 7 | 0 | 0 | 0 | Basel, Hamm, Mönchengladbach |
| Cercopithecidae | Macaca nemestrina | 29 | | 29 | 12 | 15 | 0 | 1 | 1 | Arnhem, Bali, Berlin (Tierpark), Gelsenkirchen, Osnabrück |
| Cercopithecidae | Macaca silenus | 35 | | 35 | 16 | 17 | 1 | 1 | 0 | Apeldoorn, Bekesbourne, Berlin (Zoo), Cologne, Dresden, Duisburg, Hodenhagen, Rheine |
| Cercopithecidae | Macaca sylvanus | 15 | 9†††‡‡‡ | 24 | 11 | 12 | 0 | 1 | 0 | Aachen, Rheine |

*Table 4 continued on next page*

*Table 4 continued*

| Family | Species | # Subjects tested | # Subjects drawn from literature* | Total sample | # Adult females | # Adult males | # Subadult females | # Subadult males | # Unsexed subadults | Locations for subjects in this study |
|---|---|---|---|---|---|---|---|---|---|---|
| Cercopithecidae | *Macaca tonkeana*† | | 14§§ | 14 | NA | NA | NA | NA | NA | Amsterdam, Berlin (Zoo), Dresden, Hamm, Hodenhagen |
| Cercopithecidae | *Mandrillus sphinx* | 32 | | 32 | 14 | 7 | 4 | 7 | 0 | |
| Cercopithecidae | *Papio anubis* | | 84¶¶ | 84 | 48 | 22 | 5 | 9 | 0 | Cologne, Frankfurt, Krefeld |
| Cercopithecidae | *Papio hamadryas* | 24 | | 24 | 14 | 10 | 0 | 0 | 0 | |
| Cercopithecidae | *Pygathrix cinerea* | | 18**** | 18 | 7 | 11 | 0 | 0 | 0 | |
| Cercopithecidae | *Rhinopithecus roxellana* | | 24†††† | 24 | 8 | 5 | 8 | 3 | 0 | Apeldoorn, Berlin (Zoo), Gelsenkirchen, Hannover, Heidelberg |
| Cercopithecidae | *Semnopithecus entellus* | 30 | | 30 | 17 | 7 | 4 | 2 | 0 | |
| Cercopithecidae | *Theropithecus gelada* | 38 | | 38 | 20 | 11 | 4 | 3 | 0 | Bekesbourne, Berlin (Tierpark), Magdeburg, Rheine, Stuttgart |
| Cercopithecidae | *Trachypithecus auratus* | 8 | | 8 | 3 | 0 | 3 | 2 | 0 | Bali, Stuttgart |
| Cercopithecidae | *Trachypithecus hatinhensis* | | 18**** | 18 | 8 | 10 | 0 | 0 | 0 | |
| Hominidae | *Gorilla gorilla* | | 76‡‡‡ | 76 | 22 | 18 | 19 | 17 | 0 | |
| Hominidae | *Homo sapiens* | | 127§§§ | 127 | 71 | 56 | 0 | 0 | 0 | |
| Hominidae | *Pan paniscus* | | 118‡‡‡ | 118 | 29 | 23 | 35 | 31 | 0 | |
| Hominidae | *Pan troglodytes* | | 536‡‡‡ | 536 | 186 | 138 | 110 | 102 | 0 | |
| Hominidae | *Pongo sp.* | | 47‡‡‡ | 47 | 17 | 12 | 9 | 9 | 0 | |
| Hylobatidae | *Hylobates lar* | 16 | 20¶¶¶,*****,††††† | 36 | 14 | 18 | 2 | 2 | 0 | Berlin (Tierpark), Cologne, Doué-la-Fontaine, Landau, Stuttgart, Ulm, Wuppertal |
| Hylobatidae | *Hylobates moloch* | 22 | | 22 | 8 | 5 | 4 | 5 | 0 | Bekesbourne, Lympne, Munich |
| Hylobatidae | *Nomascus gabriellae* | 6 | 4***** | 10 | 5 | 3 | 0 | 2 | 0 | Arnhem, Doué-la-Fontaine |
| Hylobatidae | *Nomascus leucogenys /siki* | 7 | 19¶¶¶,*****,‡‡‡‡‡ | 26 | 15 | 7 | 1 | 3 | 0 | Apeldoorn, Frankfurt, Osnabrück |
| Hylobatidae | *Symphalangus syndactylus* | 14 | 17¶¶¶, ***** | 31 | 12 | 11 | 4 | 4 | 0 | Arnhem, Bekesbourne, Berlin (Zoo), Dortmund, Doué-la-Fontaine, Munich, Arnhem, Hodenhagen, Osnabrück |
| Pitheciidae | *Pithecia pithecia* | 7 | | 7 | 4 | 3 | 0 | 0 | 0 | Basel, Dresden, Krefeld |
| **Total ‡** | | **501** | **1285** | **1786** | **724** | **583** | **231** | **232** | **2** | |

*Table 4 continued on next page*

Table 4 continued

| Family | Species | # Subjects tested | # Subjects drawn from literature* | Total sample | # Adult females | # Adult males | # Subadult females | # Subadult males | # Unsexed subadults | Locations for subjects in this study |
|---|---|---|---|---|---|---|---|---|---|---|
| | | | | | | | | | | |

*Fulfilling our criteria.

†Ages unknown, sex derived from given names.

‡Not including *M. tonkeana* in sex and age specific categories.

§*Nelson and Boeving, 2015*.

¶*Motes Rodrigo et al., 2018*.

**Meguerditchian et al., 2012*.

††*Phillips et al., 2007*.

‡‡*De Andrade and Sousa, 2018*.

§§*Maille et al., 2013*.

¶¶*Schweitzer et al., 2007*.

***Chatagny et al., 2013*.

†††*Schmitt et al., 2008*.

‡‡‡*Regaiolli et al., 2018*.

§§§*Canteloup et al., 2013*.

¶¶¶*Vauclair et al., 2005*.

****Cubí and Llorente, 2021*.

††††*Zhao et al., 2012*.

‡‡‡‡*Hopkins et al., 2011*.

§§§§*Cochet and Vauclair, 2012*.

¶¶¶¶*Morino et al., 2017*.

*****Caspar et al., 2018*.

†††††*Spoelstra, 2021*.

‡‡‡‡‡*Fan et al., 2017*.

the tube task. Our complete study sample with annotated respective literature sources and raw data on manual lateralization (itemized at the individual level and also at the insertion level for data generated in this study) can be viewed at Dryad (https://doi.org/10.5061/dryad.8sf7m0crv).

We classified the tested subjects into two age categories, adults (n=1307, sexually mature individuals) and subadults; the latter being comprised by infants (n=9, individuals that had not yet been weaned) and juveniles (n=456, weaned individuals that had not reached sexual maturity). If previous tube task studies assigned age categories to their subjects, we adopted this classification for the individuals concerned. In other cases and for our original data, age classification followed life history data from *Harvey and Clutton-Brock, 1985*. The taxonomy and nomenclature we apply follows *Mittermeier et al., 2013*, with the following exceptions: The recently diverging sister species pairs *Cercopithecus diana* and *Cercopithecus roloway* (Diana and Roloway monkeys) as well as *Nomascus leucogenys* and *Nomascus siki* (white-cheeked gibbons) are treated here as one respective taxonomic unit and data were pooled to obtain larger sample sizes (exact species identity of subjects is annotated in the raw data). Because the hand preference literature on orangutans (*Pongo* spp.) did not consider the species status of the individuals concerned, we analyzed respective data on the genus level. In other cases, we carefully checked the current taxonomic status of subjects drawn from the literature and tried to avoid the inclusion of interspecific hybrids. This was particularly relevant for data on lab-housed tufted capuchins (*Sapajus* spp.). If the species or hybrid status of animals was ambiguous, we did not consider them for our analyses (e.g., capuchins in *Westergaard and Suomi, 1996*).

Although available for testing at most of the institutions we visited, lemurs could not be included into the study. All tested genera (*Eulemur*, *Hapalemur*, *Propithecus*, *Varecia*) failed to manually remove food mash from the tube, despite eagerly licking it up from the ends. Only a single subject, a female *Eulemur rubriventer*, succeeded. White-faced sakis (*P. pithecia*) and Javan langurs (*Trachypithecus auratus*) were also reluctant to engage in the task, so that the final sample for these species is smaller than expected from their abundance at the institutions visited. Apart from the species considered for analysis, nine individuals belonging to miscellaneous taxa were sampled.

All experimental procedures strictly adhered to the guidance of the responsible zoo staff were approved by the institutional boards in charge and complied to the applicable animal welfare and testing regulations of the countries they were performed in. No further ethical permissions had to be obtained.

## Experimental procedure and data scoring

All species were uniformly tested in the established bimanual tube task paradigm (*Hopkins, 1995*). Due to the pronounced differences in body size between the studied species, PVC tubes of varying length and diameter were employed (*Figure 4*). Lion tamarins (*Leontopithecus*) were presented with small-sized tubes that were 5 cm long and had an inner diameter of 1 cm. Capuchins and sakis (*Sapajus*, *Pithecia*) received 10 cm × 2 cm medium-sized tubes and all remaining species large tubes measuring 10 cm × 2.5 cm. The tubes were filled with various food incentives, which differed dependent on the nutrition regimes enacted by the respective institutions. Among preferred food items for cercopithecines, gibbons, spider monkeys, and capuchins were oatmeal mixed with banana mash, soaked pellets, and boiled carrots (but note that the latter did not appeal to *Cercopithecus* and *Ateles*). Geladas (*T. gelada*) exclusively received boiled carrots. Langurs (*Semnopithecus*, *Trachypithecus*) and sakis were preferably tested with boiled rice, and tubes for the latter were also stowed with nuts as an additional incentive. Lion tamarins received tubes filled with pure banana mash or commercial tamarin pie. Primates were preferably tested within their social groups. A separation of individuals was only undertaken in exceptional cases when it was necessary to counteract social tension created by the presentation of the tubes. Dependent on the constructional restraints of the enclosures, tubes were either placed into a separated part of the enclosure before the primates could enter or were handed over directly through the wire mesh. In the latter case, the hand that the experimenter used to offer the tube was noted. To check whether the hand used by the experimenter to offer the tube had an effect on the directional hand preferences of the tested primates, we ran a linear mixed effect model employing a binomial link function. No effect on the recovered hand preferences in the respective sessions was found (t=–1.31, SE = 0.02, p=0.191).

The tube tasks were recorded with digital cameras and scored from the video footage. For each subject, we obtained a minimum of 30 bimanual insertions (one hand is holding the tube, the other

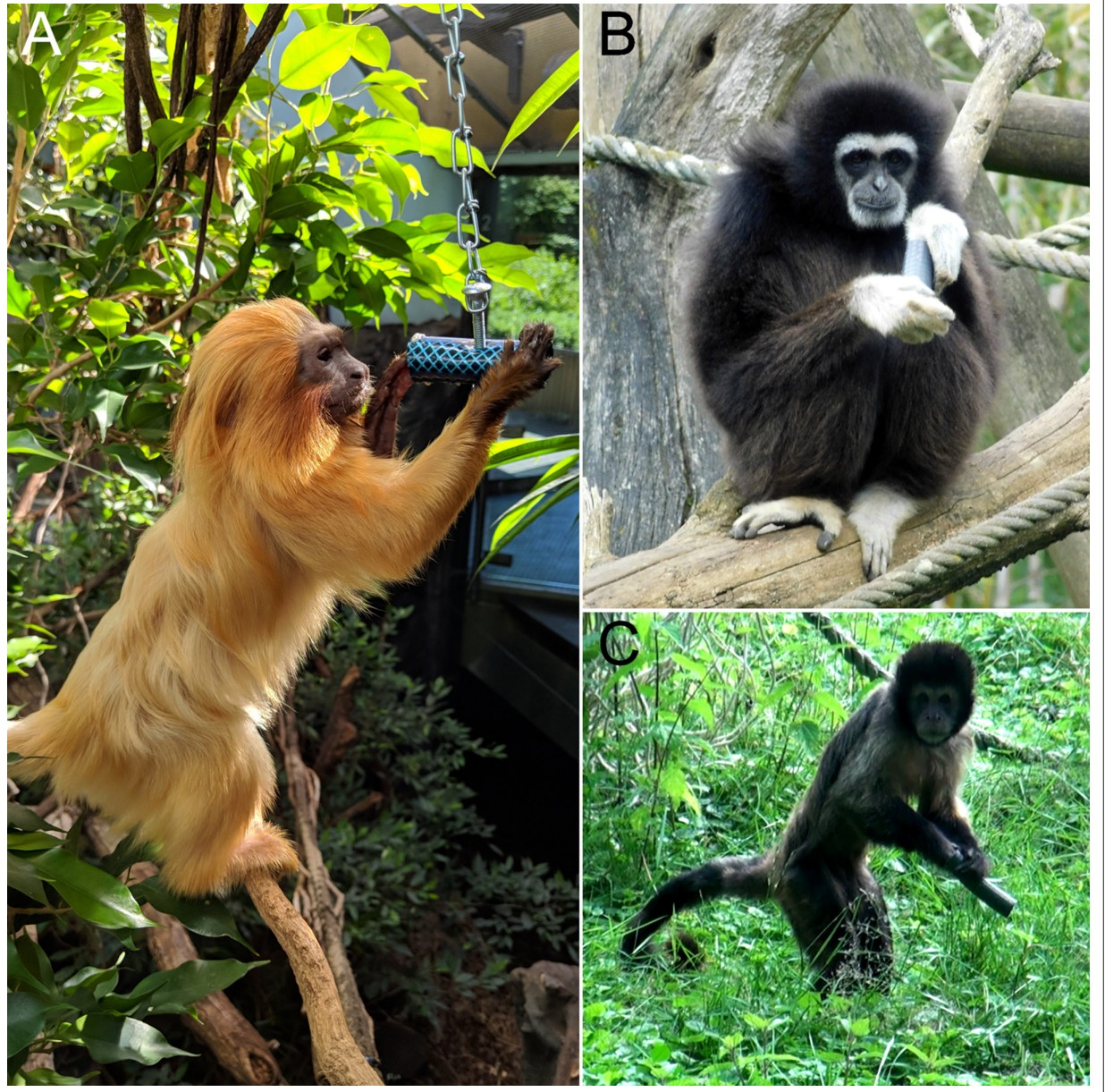

**Figure 4.** Various anthropoid primates engaging in the tube task. (**A**) Golden lion tamarin (*Leontopithecus rosalia*) manipulating a small tube at Zoo Frankfurt, Germany. (**B**) White-handed gibbon (*Hylobates lar*) handling a large tube at Bioparc de Doué-la-Fontaine, France. Note that the thumb is used to probe into the tube, an insertion pattern characteristic of gibbons. (**C**) Yellow-breasted capuchin (*Sapajus xanthosternos*) engaging in the task with a medium-sized tube while adopting an erect bipedal stance at ZooParc Overloon, the Netherlands. Photographs by Kai R Caspar.

one is retrieving food; mean number of insertions: 50.66±20.98, range: 30–155) in at least six bouts (uninterrupted manipulation sequences, as defined by *Morino et al., 2017*; mean number of bouts: 20.60±11.13, range: 6–82). Literature data for individual subjects had to match or exceed these thresholds to be included into the analysis. Unimanual or foot-assisted insertions were not scored and were, whenever possible, also carefully excluded from the literature data. We noted the digit used to extract the food as well as the body posture of the manipulating individual (sitting, crouched bipedal stance, erect bipedal stance, suspended [clinging to wire mesh or other substrates without the hands stabilizing posture, always tail-assisted in spider monkeys]). The vast majority of responses were observed in a sitting position (n=22,993; 90.6%). Due to this imbalance, because posture-related information was mostly unavailable for literature data, and since its influence on manual laterality already received great research attention in the past (*Sanford et al., 1984*; *Westergaard et al., 1998*;

*Blois-Heulin et al., 2007*; *Laurence et al., 2011*), we refrained from including posture effects into our analyses. Nevertheless, for potential future use by other researchers, we decided to include this measure, alongside information on digit use ('N.A.' if ambiguous in the respective footage) during manipulation, alongside the respective raw data.

## Statistics

Data were analyzed in R (*R Development Core Team, 2020*). Preferably, analyses were performed on insertion data (also called frequencies) instead of manipulations bouts to approximate laterality. Unfortunately, not all available tube task studies provided insertion data (e.g., *Maille et al., 2013*; *Fan et al., 2017*; *Spoelstra, 2021*), so that in the final dataset, estimates of manual lateralization based on insertions and bouts had to be mixed for certain species (*Cercocebus torquatus*, *Cercopithecus neglectus*, *Hylobates lar*, *N. leucogenys*, and *Sapajus* spp.). However, since previous work demonstrated that hand preferences derived from bouts and insertions are highly correlated and non-conflicting, we do not consider this a confounding factor for our analysis (*Hopkins et al., 2001*; *Hopkins, 2013b*).

For quantifying lateralized responses on the individual level, we calculated HIs for all subjects as well as the corresponding binomial z scores to allow grouping into hand preference categories. HI is a descriptive index that can range from –1 (all manipulations left-handed) to 1 (all right-handed) and is calculated via the formula HI = (R − L)/(R + L). The z score, on the other hand, indicates whether there is a statistically significant bias in hand use. Following established criteria (*Hopkins, 2013b*), we rated subjects with z score values higher than 1.96 as right-handed, those with values lower than −1.96 as left-handed, and the remaining ones as ambipreferent. We use the term 'ambipreferent' here instead of 'ambidextrous' or 'ambiguously handed' to indicate a lack of preferences, because the latter two expressions have clear non-synonymous definitions when applied to humans, but are not consistently used in the non-human primate literature (*Hopkins et al., 2013a*). At the species level, we used the mean HI of subjects as a measure of lateralization direction and the mean of absolute HI values (MeanAbsHI) as a measure of strength.

We applied one-sample t-tests to each species and genus sample encompassing data for at least 15 individuals to check whether HI distributions were significantly skewed at the population level. Additionally, the chi-square goodness-of-fit test was employed to test if the numbers of left- and right-handers as well as ambipreferent individuals differed from a baseline distribution. Earlier studies performed the goodness-of-fit test against the null hypothesis of a chance distribution of the three hand preference categories (*Vauclair et al., 2005*). Due to our large multispecies sample, we could adopt a different approach: For each of the three major clades studied (Cercopithecoidea, Hominoidea, and Platyrrhini), we calculated the mean frequencies of individuals being either ambipreferent or handed. We then assumed an equal probability of handed individuals to be either right- or left-handed and this way calculated a clade-specific baseline. Distributions for each species were then compared to this clade-specific average. For the Hominoidea, we excluded humans for the calculation of the baseline, because their evidently extreme lateralization bias would have otherwise skewed the results. Bonferroni correction was employed to address multiple testing.

We employed the phytools package version 0.7–70 (*Revell, 2012*) to visualize evolutionary patterns, quantify phylogenetic signals (by employing Pagel's $\lambda$ – *Freckleton et al., 2002*; the null hypothesis of no phylogenetic signal was tested by means of the likelihood ratio test), and to calculate maximum likelihood ancestral state estimates (*Supplementary file 1*), each separately for direction and strength of lateralization. Time-calibrated primate phylogenies were derived from the 10kTrees website (*Arnold et al., 2010*). Three species in our study were not included in the respective database, *Ateles hybridus*, *Sapajus flavius*, and *Trachypithecus hatinhensis*. We therefore replaced *A. hybridus* and *T. hatinhensis* in the tree with respective sister taxa, namely *A. belzebuth* and *T. francoisi* (*Morales-Jimenez et al., 2015*; *Roos et al., 2019*), for which data were provided. This way the topology and branch lengths of the tree could be kept equal to a model which would have included the actual species we studied. The same could not be done for *S. flavius*, so that it was amended manually in the respective trees by relying on divergence dating from *Lima et al., 2018*.

We computed PGLS regression models to test the effects of different biological variables on hand preferences while addressing phylogeny (correlation structure: Pagel's $\lambda$ ; model fit: maximum likelihood). The R packages *ape* (*Paradis et al., 2019*), *nlme* (*Pinheiro, 2020*), and *MuMIn* (*Barton,*

*2020*) were used for model creation and evaluation. We used multi-model inference to test how well hypothesis-derived predictors could explain HI measures on the species level. Predictor-based models were ranked against a NULL model to estimate their explanatory power and identify the best-performing one (*dredge* function in *MuMIn*). Second-order Akaike information criteria (AICc) and respective Akaike weights were used to evaluate model components. The normality of model residuals was checked by applying the Shapiro-Wilk test. We relied on the conditional model average to assess effects of individual predictors. We selected the following variables as model predictors to address established hypotheses on the evolution on primate hand preferences: Ecology (terrestrial vs. arboreal), occurrence of habitual foraging-related tool use in the wild (binarily coded), and endo-cranial volume (numeric, log-transformed) of females (see *Supplementary file 3* for predictor data and respective references). A potential multicollinearity of predictors was checked by computing their variable inflation factors. We ran the models on the species means for HI and AbsHI, respectively, resulting in two separate analyses for direction and strength of population-level laterality. All species (n=38) were included into the strength (AbsHI) analysis. For direction, we only considered species for which we had at least 15 sampled individuals, resulting in a more restricted sample (n=34). Since humans are extreme outliers in regard to their hand preference direction, we decided to compute a second direction model to identify potential biases that might derive from their inclusion. Thus, this second model on hand preference direction encompassed a species sample of n=33.

Finally, we employed Bayesian phylogenetic multilevel models by aid of the *brms* package (*Bürkner, 2017*) to check whether individual HI and AbsHI were influenced by age and sex (excluding unsexed individuals; n=2). Analyses were run for the complete sample as well as within the three superordinate clades. This resulted in a total of eight models, four for each of the two response variables. We assumed a notable effect on lateralization patterns when the model's 95% credible intervals of intercept and respective regression coefficients did not overlap with zero. Default priors were used. Modelling results including the number of chains and iterations performed are summarized and visualized in *Supplementary file 2*. Due to their highly derived handedness patterns, we again excluded humans from these analyses to avoid skewing the results. To further avoid bias, we also removed chimpanzees from the models, since they are vastly overrepresented in our sample (31% of total individuals and 47% of all subadults). Finally, as no age data on Tonkean macaques (*Macaca tonkeana*) were available, this species was not featured in these analyses as well. This left us with n=1107 individuals in the full dataset, n=366 in the hominoid, n=459 in the cercopithecoid, and n=282 in the platyrrhine subsample.

## Acknowledgements

We would like to thank the curators and animal keepers at the following institutions for the permission to study their animals and for their support during data collection:

Allwetterzoo Münster, Apenheul (Apeldoorn), Artis Amsterdam, Bali Wildlife Rescue Center, Bioparc de Doué-la-Fontaine, Burger's Zoo (Arnhem), Erlebnis-Zoo Hannover, Howletts Wild Animal Park (Bekesbourne), Naturzoo Rheine, Parc Zoologique et Botanique de Mulhouse, Port Lympne Wild Animal Park (Lympne), Serengeti-Park Hodenhagen, Tierpark Aachen, Tierpark Berlin, Tierpark Hamm, Tierpark Hellabrunn (Munich), Tierpark Mönchengladbach, Tierpark Ulm, Wilhelma Stuttgart, Zoo Basel, Zoo Berlin, Zoo Dortmund, Zoo Dresden, Zoo Duisburg, Zoo Frankfurt, Zoo Heidelberg, Zoo Köln (Cologne), Zoo Krefeld, Zoo Landau in der Pfalz, Zoo Liberec, Zoo Magdeburg, Zoo Neuwied, Zoo Osnabrück, Zoo Wuppertal, Zoológico de São Paulo, Zoologischer Stadtgarten Karlsruhe, ZOOM Erlebniswelt Gelsenkirchen, ZooParc Overloon, and Zoopark Erfurt.

We are also indebted to Hélène Cochet, Julia Fischer, William D Hopkins, Adrien Meguerditchian, and Luca Morino for sharing raw data on bimanual task preferences in humans, barbary macaques, great apes, olive baboons, and gibbons, respectively.

Daniel Issel is greatly acknowledged for collecting data on pig-tailed macaques and Javan langurs in Bali and Petra Bolechová for doing so for Diana monkeys at Zoo Liberec. Finally, we would like to thank Miriam Lindenmeier and Larissa Günther for crucial assistance during data collection at various locations and Alex DeCasien as well as Adrien Meguerditchian for their constructive and insightful reviews of earlier versions of this manuscript.

This research was funded by the German Society for Mammalian Biology (Deutsche Gesellschaft für Säugetierkunde). KRC was supported by a PhD fellowship of the German Academic Scholarship Foundation (Studienstiftung des deutschen Volkes e.V.).

## Additional information

### Funding

| Funder | Grant reference number | Author |
|---|---|---|
| German Society for Mammalian Biology | | Kai R Caspar |
| Studienstiftung des Deutschen Volkes | | Kai R Caspar |

The funders had no role in study design, data collection and interpretation, or the decision to submit the work for publication.

### Author contributions

Kai R Caspar, Conceptualization, Data curation, Formal analysis, Funding acquisition, Investigation, Visualization, Methodology, Writing – original draft, Project administration, Writing – review and editing; Fabian Pallasdies, Data curation, Formal analysis, Investigation, Visualization, Methodology, Writing – review and editing; Larissa Mader, Heitor Sartorelli, Investigation, Writing – review and editing; Sabine Begall, Data curation, Formal analysis, Supervision, Investigation, Methodology, Writing – review and editing

### Author ORCIDs

Kai R Caspar  http://orcid.org/0000-0002-2112-1050
Fabian Pallasdies  http://orcid.org/0000-0001-5359-4699
Sabine Begall  http://orcid.org/0000-0001-9907-6387

### Ethics

All animals participated voluntarily in the behavioral tasks studied. The experiments were approved by and strictly adhered to the guidance of the staff in charge. All experimental procedures complied to the applicable animal welfare and testing regulations. No further ethical permissions had to be obtained. In a judicial sense, none of the experiments represented "animal testing" in the countries that they were performed in.

### Decision letter and Author response

Decision letter https://doi.org/10.7554/eLife.77875.sa1
Author response https://doi.org/10.7554/eLife.77875.sa2

## Additional files

### Supplementary files

• Supplementary file 1. Estimates of ancestral manual lateralization patterns in anthropoid taxa.

• Supplementary file 2. Results and visualization of Bayesian models to infer effects of sex and age cohort on individual-level hand preference patterns.

• Supplementary file 3. Species-level predictors used for phylogenetic generalized least squares (PGLS) modelling.

• Transparent reporting form

### Data availability

All data generated or analysed during this study have been deposited on Dryad: https://doi.org/10.5061/dryad.8sf7m0crv R code relevant to statistical analyses is deposited here: https://github.com/fpallasdies/CasparEtAl2022PrimateHandPreference.git, (copy archived at swh:1:rev:69cb316acaf82b1650386b5d8fcc9964e611ca01).

The following dataset was generated:

| Author(s) | Year | Dataset title | Dataset URL | Database and Identifier |
|---|---|---|---|---|
| Caspar KR, Pallasdies F, Mader L, Sartorelli H, Begall S | 2022 | The evolution and biological correlates of hand preferences in anthropoid primates | https://doi.org/10.5061/dryad.8sf7m0crv | Dryad Digital Repository, 10.5061/dryad.8sf7m0crv |

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
