## [Editor Report]

This paper combines new and previously generated data on hand preference to show that hand preference strength, but not direction, is predicted by ecology and phylogeny across primates. By drawing on the most expansive data set to date on experimentally determined hand preference, it calls existing hypotheses on the evolution of hand preference into question and shows that the strength of lateralization in humans is unusually extreme. Its results are of interest for evolutionary anthropologists, primatologists, and morphologists interested in the evolution of lateralization and human uniqueness.

---

## [Decision Letter]

**Decision letter after peer review:**

Thank you for submitting your article "The evolution and biological correlates of hand preferences in anthropoid primates" for consideration by *eLife*. Your article has been reviewed by 2 peer reviewers, and the evaluation has been overseen by a Reviewing Editor and Molly Przeworski as the Senior Editor. The following individuals involved in the review of your submission have agreed to reveal their identity: Alex DeCasien (Reviewer #1); Adrien Meguerditchian (Reviewer #2).

Essential revisions:

1) Control for phylogenetic structure in individual-level analyses of the effects of age, sex, etc. on handedness; Reviewer 1 suggests specific methods that should allow you to preserve individual-level data in a phylogenetically controlled framework.

2) Consider Reviewer 2's concerns that the absence of an arboreality/terrestriality signal for handedness may in part be due to noisy estimates of handedness in species with small sample sizes. Possible approaches include moderating your claims and discussing how uneven sampling affects your conclusions, or analyses that provide bounds on how large a true effect size could be to remain consistent with your observations.

3) Evaluate the robustness of your findings to alternative species-level classifications of arboreality/terrestriality.

*Reviewer #1 (Recommendations for the authors):*

I suggest publication following major revisions, mainly due to a lack of phylogenetic modelling when testing for sex, age, and subgroup effects on handedness. Below, my feedback is organized by section.

Abstract:

– The abstract is clear and well-written.

Introduction:

– The beginning of the introduction states as a fact the uniqueness of human handedness, but one goal of the manuscript is to test exactly how unique this is among anthropoids. I suggest rewording to address this (i.e., the uncertainty and gaps in the literature that are addressed by the analyses in this manuscript).

– Lines 65-68: Perhaps I am misunderstanding, but it is not immediately clear how a population-level right-hand bias can exist while the proportion of right-handers is ~50% -- can the authors clarify here?

– Lines 71-91: Can the authors expand upon why the hypotheses rely on the left versus right-handedness for certain skills? It is not clear, for example, why terrestrial lineages would evolve any hand preference at all if they are not constrained by posture stabilization.

– Lines 92-105: I suggest separating the discussion of tool use and brain size hypotheses since they speak to different mechanisms (ultimate vs. proximate).

Methods:

– The data collection methods are detailed and thoughtfully described.

– Relevant to results discussed in Lines 364-376: For models of sex, age, and subgroup effects, individual-level data can be tested in a phylogenetic framework using e.g., Bayesian phylogenetic generalized linear mixed models in the R package MCMCglmm (Hadfield 2010). Such analyses would be more appropriate than the non-phylogenetic methods implemented here.

– Please demonstrate that the assumptions of linear modelling were tested and confirmed (e.g., normality of residuals) to justify a lack of data transformations (e.g., log-transformation). Also, log-transformation of brain size (from Results Tables) should be mentioned and justified here. In addition, please test predictor multicollinearity and report VIFs.

Results:

– The results are clearly stated, and the figures are comprehensive and visually appealing.

– Figure 2: Labelling 0 on the MeanHI legend would improve interpretability.

– Figure 4: Please add information regarding the directionality of the ecology estimate.

Discussion:

– The discussion is thorough and clear.

I hope the authors find these comments useful in revising their manuscript.

*Reviewer #2 (Recommendations for the authors):*

Here are my main comments and questions that I hope to be constructive to a further revision of the manuscript:

(1) I believe this large review and study is very important and relevant although I remain skeptical with the strong claim (supported by the lake of findings resulting from the quantitative phylogenetic methods) that the dichotomy of arboreal versus terrestrial lifestyle has nothing to do with the direction of population-level handedness in a non-human primate. I believe it is not excluded that the results might remain equivocal at the species level for the following reasons. In fact, the question of the lack of statistical power at the species level (related to the poor sample size of subjects in most species) is not addressed although it might constitute a serious limitation for concluding a lack of correlation with the selected biological predictors at the species-level. For instance, after compiling HI data in nonhuman primates (humans are always excluded in the next analyses) not at the species level but rather at the clade level to increase statistical power and to go beyond this limitation, a quick comparison of Mean HI between arboreal versus terrestrial species and simple t-test shows a significant difference, the terrestrial primates being significantly more rightward than arboreal primates (or the arboreal primates being significantly leftward than the terrestrial).

This is true not only in Hominoidea (see Author response image 1 even after excluding the chimpanzees' data, see Author response image 1, the highly represented species) but also in Cercopithecoidea (see Author response image 1), and it is thus true after compiling all clades together even the non-significant Platyrrhini (see Author response image 1).

**Decision letter image 1. sa1fig1:** 

Although those differences are certainly driven by some overrepresented species, the question of sample size may be critical given most of the significant results at the species level (and consistent with the "novel postural origin hypothesis") seem to come from studies involving a relatively high sample size (such as chimpanzees, gorillas, baboons, etc.).

Even if "species-level direction of manual lateralization was largely uniform among non-human primates and neither correlated with phylogeny nor with any of the selected biological predictors" according to the quantitative phylogenetic methods used by the authors, such a consistent and significant direction of the effect is not at the species level but at the clade levels across those two different clades raises some questions (if not two consecutive false positive or driven by cofounding factors other than ecology). Given those differences are driven by some overrepresented species, does it mean that effect is significant only when considering species with high sample size and robust statistical power? Does it then mean that lack of effect at the species level might be attributed to low sample sizes in most of the species? Regarding the highly heterogenous sample size across species and effects found at the clade level, the authors should address those questions and such a statistical power issue which could make the results at the species-level equivocal regarding the hypothesis tested in this study.

(2) Lines 268-269: "To prevent these taxa to skew the comparisons, we restricted their respective sample sizes to the numbers of individuals in the second-largest species sample for their respective clade."

To be honest, although I understand this typical issue (that highly represented species naturally skew the results), the issue could be seen the other way around (as mentioned in the previous comment 1): Considering the hypothesis that a high sample size is critical at the species-level to evaluate and infer population-level handedness in nonhuman primates, it is particularly unfair and strange to arbitrary restricted the sample-size and weaken the only data-sets that are relevant and significant because of their statistical power. After such a generalization of the weaknesses of the sample sizes even to the species with strong evidence, the lack of effects of the selected biological predictors at the species-level as well as the uniformity of the species-level direction of manual lateralization reported among non-human primates remains then questionable.

If the analyses are made from the individual HI scores, how such a restriction has been done? Which individual HI scores in the chimpanzees and in the olive baboons have been kept in the analyses and which ones have been excluded?

(3) How terrestrial versus arboreal lifestyle has been attributed to species? Especially macaques? Nemestrina are usually classified as terrestrial while fascicularis and Silenus as arboreal (sylvanus and Tonkeana as both terrestrial and arboreal)?

Some species seem to have been attributed to the wrong category. For instance, *Cercocebus torquatus*, Cercopithecus diana, *Cercopithecus neglectus* are usually classified as mainly arboreal (although some of them could also move on the ground) but could not be considered as mainly terrestrial.

(4) Lines 483-484: "it is obvious that other hand use situations, such as gesturing, need to be considered to arrive at a holistic understanding of primate hand use evolution"

Do you mean "communicative gesturing"? If so, a reference might be useful here.

(5) I'll be happy to share our baboon *Papio anubis* data set (Vauclair et al. 2005) from which the HI scores have been not included in the meta-analysis.

Congratulations again for this amazing and considerable work. I had such a pleasure reading it and hope my comments and questions were useful.

--

HOMINOIDEA (Author response image 1)

Arbo

M.HI = – 0.10000000000

SE = 0.05021825800

N = 172

Terrest

M.HI = 0.13024938100

SE = 0.02151882000

N = 730

t-test comparison A/B : t(900) = 4.5516, p < 0.0001

--

HOMINOIDEA without the chimps (Author response image 1)

Arbo

M.HI = – 0.10000000000

SE = 0.05021825800

N = 172

Terrest

M.HI = 0.123932767

SE = 0.043371804

N = 193

t-test comparison A/B : t (364) = 3.3923, p = 0.0008

--

CERCOPITHECOIDEA (Author response image 1)

Arbo

Mean = -0.07950534100

SEM = 0.04680844900

N = 237

Terrest

M.HI = 0.054

SE = 0.03505

N = 257

t-test comparison A/B : t (492) = 2.3057, p = 0.0215

--

TOTAL (Author response image 1)

Arboreal

M.HI = -0.056987408 S.E. = 0.027564288 N = 692

t(691) = 2.0674, p = 0.0391

Terrestrial

M.HI = 0.112683704 S.E. = 0.018593258 N = 958

t(957) = 6.0605, p<0.0001

t-test comparison A/B : t (1648) = 5.2935, p<0.0001

[Editors’ note: further revisions were suggested prior to acceptance, as described below.]

Thank you for resubmitting your work entitled "The evolution and biological correlates of hand preferences in anthropoid primates" for further consideration by *eLife*. Your revised article has been evaluated by Molly Przeworski (Senior Editor) and a Reviewing Editor.

The manuscript is responsive to previous reviewer concerns and is near-ready for publication; congratulations on a nice revision! We are returning it to you in the event you would like to address the remaining interpretive concerns raised in Reviewer 2's comments below. Specifically:

1. Reviewer 2 makes an argument that the paper is overly dismissive of the (novel) POH hypothesis, given weak evidence for a potential role of terrestrial versus arboreal ecology in lateralization direction (in the predicted direction). The reviewer also asks for more discussion of whether sample size/power issues contribute to weak support for existing hypotheses.

Please consider whether you would like to moderate language in your abstract or discussion in light of these arguments (the Reviewer points to some specific examples). Ultimately, the interpretation of your results is not extreme and is your prerogative, so we do not consider this a prerequisite for publication. However, it may be premature to base support or rejection of a long-standing hypothesis based on p-values of 0.04 versus 0.07, which have a very similar level of evidentiary strength (or weakness).

*Reviewer #2 (Recommendations for the authors):*

Thank you for the revision of your manuscript. I am glad that the authors finally acknowledged and reported a significant effect for the ecological signal (arboreal versus terrestrial ecology) in the direction data, which, to me, change significantly the initial straightforward conclusions of the paper. But I am afraid that the authors preferred to maintain their initial firm and unbalanced conclusion after downplaying the potential relevance of this effect. I am also satisfied by the responses and justifications provided in the comments 2 and 3.

Here are my remaining concerns.

1) Ecological factor (arboreal versus terrestrial ecology) is presented in the paper as one of the few key biological factors and hypothesis that are willing to be tested in the present study. It turned out, that, in the present revision after my first round of comments, the authors finally did find a significant result (p=0.04) for this ecological signal in the direction data (which still reaches conventional level of significance at p<0.07 even after removing the human species). To be honest, I found thus quite surprising, partial and unfair that this result is poorly discussed, not even mentioned in the abstract but rather even just quickly downplayed in the discussion with any justification, other than "the weakness of the effect" and other than a laconic claim (at lines 307-311) : "Admittedly, improved sampling, both within and across species might consolidate this pattern in the future, so that we do not want to dismiss the POH at this point. However, even if its predictions should become validated eventually, the hypothesis does still not provide an evolutionary mechanism for why population-level hand preference patterns should be coupled to positional behavior."

This is unfortunate and a bit short as a discussion after such a flipping finding that might be consistent with the "novel PO hypothesis". Despite evidence, the authors preferred ignoring interpreting this potential effect (other than downplaying it) and give thus the feeling of not wanting to contradict their initial firm conclusion and thus kept only one side of the potential interpretations, namely for instance:

– "Species-level direction of manual lateralization was largely uniform among non-human primates and did not notably correlate with any of the selected biological predictors" (lines 19-21).

– "Furthermore, the evolutionary scenario proposed by both versions of the POH is outdated and should not be perpetuated without explicitly stating its shortcomings." (lines 313-314)

– "no phylogenetic or ecological signal was found for lateralization direction, and few species exhibit significant population-level hand preferences. We tested (…) the POH (…) hypotheses, but none were corroborated by our data." (lines 435-438)

I totally agree that given (1) the effect is not strong, not at p<0.0001, but rather at 0.04 and (2) even decreased at p=0.07 when excluding humans, and that (3) we should indeed never exclude, in behavioral science, a potential false positive effect, (4) and that further investigation would help keep testing this small effect by increasing statistical power for each species. So any firm conclusion should not be drawn from such a small effect, let alone the firm conclusion of the authors ignoring this effect. Nevertheless, I recommend the authors to, at least, try to interpret this potential effect which seems congruent with the "novel PO hypothesis" by presenting this other side of the potential interpretations, not just one side that does fit with their initial conclusion.

A more balanced view including both opposite sides of the interpretation of this effect would be thus more fair and impartial (i.e. "congruent with the "novel PO hypothesis" versus "congruent with the initial conclusion of the paper").

Moreover, when I played with the data analyses (and likely with the grey zone of attribution of "arboreal" versus "terrestrial" categorial attribution for some ambiguous species, see previous initial comment 2 and its response), I found a stronger and significant difference between the mean HIs of terrestrial versus arboreal non-humans (terrestrial : M.HI = 0.11, SE = 0.02; arboreal: M.HI = -0.06, SE = 0.03) in comparison to the authors with their justified classification (MeanHIarboreal: -0.08, SD: 0.16; MeanHIterrestrial: 0.04, SD: 0.10).

Bonus comment (for the pleasure of the discussion):

Maybe one evolutionary hypothesis to try interpreting the effect in accordance with the "novel PO" might be that, in the primate evolution, a tendency to predominance of right-handedness would be inherited from a the common ancestor between Catarrhini and Hominid (but not from the Platyrrhini). In that case, this right-bias would be thus mostly altered in arboreal ecology (given its implication in supporting the body in the trees) whereas more visible in terrestrial ecology without such a body postural constraint; a right predominance bias that would dramatically increase in the course of the evolution in the bidedal *Homo sapiens*, the most terrestrial primates (but also to a lesser extend the gorillas, the second most terrestrial primates).

So, as mentioned by the authors, only further investigations by increasing the sample size per species (and the number of species) would help disambiguating those two opposite sides, but for that, the two sides should be fairly considered in the paper…

2) Lines 255-257: "It is therefore premature to assume that hominids display qualitatively different population-level lateralization patterns than other primates."

Lines 268-270: "Again, we suggest that an expanded and more balanced dataset could level out the lateralization differences between (great) apes and other anthropoids."

I am not sure to understand why? Does it mean that increasing the statistical power in other non-hominid species (such as the significantly right-handed baboon *Papio anubis*, at N = 104) to reach the sample size of great apes would likely make them converge toward more significant biases at the populational-level? If so, it would be very interesting for getting the full picture of primate handedness evolution.

Or in contrary, decrease the statistical power in great apes would make them non-significantly lateralized (but less representative), just like most of the non-hominid primates?

In both cases, the rational of those both alternatives are not entirely clear to me.

I would rather suggest the authors to provide, as I initially suggested in my first review, a proper and full discussion about the issue of sample size that occurred in the primate handedness literature, and its minimum threshold to be representative enough for inferring and comparing direction and degree of population-level handedness among nonhuman primates.

In fact, given the weaker degree of population-level handedness bias in nonhuman primates in comparison to humans, increasing sample size of subjects in nonhuman primates is thus an important consideration for being able to detect representative directional and degree of asymmetries at population-level and requires thus the largest sample size of subjects whenever possible (see the statistical demonstration in Hopkins and Cantalupo, 2005, and see Hopkins 2006, recommending more than 50 subjects per species).

Only 5 species in the literature reviewed in the present paper reached that sample-size threshold of 50 subjects and that level of statistical power, and actually…all of them showed significant population-level handedness!

Gorillas, Gorilla gorilla, N = 76 => significant right-handedness

Adult Bonobos, Pan Paniscus, N = 52 adults => significant right-handedness

Chimpanzees, *Pan troglodytes*, N = 536 => significant right-handedness

Baboons, *Papio anubis*, N = 104 => significant right-handedness

Humans, *Homo sapiens*, N = 127 => significant right-handedness

Even after downing the sample-size threshold to 40 subjects, out of the 7 species which reached that statistical power, only 1 species, the *Ateles* fuscicepsall (N=46), does not show any populational bias, while the Orangutans, Pongo sp., N = 47 => significant left-handedness.

So one of the main limitations of the nonhuman primate literature (including the present study) on both handedness and brain asymmetries is the lack of sample size which prevent doing robust comparative analyses at the species-level with humans or great apes. With such a lack of statistical power in 33 species (out of 38) which are below this sample size threshold, it should not be excluded that we may have to deal with not representative and inconsistent findings (e.g., strong bias in some monkeys, lack of populational bias in other, etc.), which could not be completely covered and addressed by clade-level analyses. This is a discussion point that should be better addressed.

3) At table 1, the data and analysis presented from our baboons' study *Papio anubis* (N=104), are not accurate and might affect the general analyses of the paper. Only 84 subjects are mentioned, instead of N = 104, and the value of the M.HI, presented inaccurately as non-significant (p=0.102), does not correspond. In our results, the M.HI was significant (p<0.05) and was rather M.HI = 0.13, SE = 0.06, N=104 (see the updated data file that I sent including the *Papio anubis* data).

Moreover, this study is the only one in monkeys approaching the sample size and statistical power of most great apes' studies, converging with significant populational bias finding. I believe this finding should be better integrated within the discussion made by the authors about significant findings in large sample size in great apes only, as well as the discussion about the "sample size issues in handedness NH primate studies" requested in the previous comment.

Overall, despite my concerns, I insist that this study is very important for the field (that is why I believe the conclusions should be thus more balanced) and I renew my congratulations on this amazing and considerable work. I had such a pleasure reading it and hope my comments and questions were useful.

---

## [Author Response]

Reviewer #1 (Recommendations for the authors):I suggest publication following major revisions, mainly due to a lack of phylogenetic modelling when testing for sex, age, and subgroup effects on handedness. Below, my feedback is organized by section.Abstract:– The abstract is clear and well-written.Introduction:– The beginning of the introduction states as a fact the uniqueness of human handedness, but one goal of the manuscript is to test exactly how unique this is among anthropoids. I suggest rewording to address this (i.e., the uncertainty and gaps in the literature that are addressed by the analyses in this manuscript).

It was not our intention to phrase this as a fact, but we tried to indicate that given all the so far available data, humans appear to be unique. We have adapted the wording in the paragraph for greater clarity and to further reflect uncertainty.

– Lines 65-68: Perhaps I am misunderstanding, but it is not immediately clear how a population-level right-hand bias can exist while the proportion of right-handers is ~50% -- can the authors clarify here?

A significant skew with 50% right-handers in a population can be achieved if there is a large fraction of ambipreferent and only few left-handed individuals present, as is the case in the respective primate species. We now mention this in the text.

– Lines 71-91: Can the authors expand upon why the hypotheses rely on the left versus right-handedness for certain skills? It is not clear, for example, why terrestrial lineages would evolve any hand preference at all if they are not constrained by posture stabilization.

Unfortunately, no mechanistic explanation for this has been suggested so far. This idea is solely based on correlational evidence from hand preference studies in different primate clades. We added a sentence clarifying this issue to the respective section (l. 95 f). That is of course a major weakness of the hypothesis but we feel that it goes beyond the scope of our paper to rationalize the POH, in particular because we call for abandoning/reforming it later on.

– Lines 92-105: I suggest separating the discussion of tool use and brain size hypotheses since they speak to different mechanisms (ultimate vs. proximate).

We are not sure what the reviewer suggests us to adapt here. Both hypotheses present ultimate evolutionary scenarios.

Methods:– The data collection methods are detailed and thoughtfully described.– Relevant to results discussed in Lines 364-376: For models of sex, age, and subgroup effects, individual-level data can be tested in a phylogenetic framework using e.g., Bayesian phylogenetic generalized linear mixed models in the R package MCMCglmm (Hadfield 2010). Such analyses would be more appropriate than the non-phylogenetic methods implemented here.

We followed your recommendation and have now implemented Bayesian phylogenetic multilevel models to replace the Wilcoxon tests from the previous manuscript version. Results are presented and visualized in Supplementary File 1.

– Please demonstrate that the assumptions of linear modelling were tested and confirmed (e.g., normality of residuals) to justify a lack of data transformations (e.g., log-transformation). Also, log-transformation of brain size (from Results Tables) should be mentioned and justified here. In addition, please test predictor multicollinearity and report VIFs.

It is now explicitly mentioned that the normality of residuals was checked (l. 297) as well as that logtransformation was applied to the endocranial volume data (l. 301) in the Methods section and present the VIFs of all predictors in Table 3 within the Results section.

Results:– The results are clearly stated, and the figures are comprehensive and visually appealing.– Figure 2: Labelling 0 on the MeanHI legend would improve interpretability.

Adjusted

– Figure 4: Please add information regarding the directionality of the ecology estimate.

Adjusted

Discussion:– The discussion is thorough and clear.

Thank you!

Reviewer #2 (Recommendations for the authors):Here are my main comments and questions that I hope to be constructive to a further revision of the manuscript:(1) I believe this large review and study is very important and relevant although I remain skeptical with the strong claim (supported by the lake of findings resulting from the quantitative phylogenetic methods) that the dichotomy of arboreal versus terrestrial lifestyle has nothing to do with the direction of population-level handedness in a non-human primate. I believe it is not excluded that the results might remain equivocal at the species level for the following reasons. In fact, the question of the lack of statistical power at the species level (related to the poor sample size of subjects in most species) is not addressed although it might constitute a serious limitation for concluding a lack of correlation with the selected biological predictors at the species-level. For instance, after compiling HI data in nonhuman primates (humans are always excluded in the next analyses) not at the species level but rather at the clade level to increase statistical power and to go beyond this limitation, a quick comparison of Mean HI between arboreal versus terrestrial species and simple t-test shows a significant difference, the terrestrial primates being significantly more rightward than arboreal primates (or the arboreal primates being significantly leftward than the terrestrial).This is true not only in Hominoidea (see Author response image 1 even after excluding the chimpanzees' data, see Author response image 1, the highly represented species) but also in Cercopithecoidea (see Author response image 1), and it is thus true after compiling all clades together even the non-significant Platyrrhini (see Author response image 1).Although those differences are certainly driven by some overrepresented species, the question of sample size may be critical given most of the significant results at the species level (and consistent with the "novel postural origin hypothesis") seem to come from studies involving a relatively high sample size (such as chimpanzees, gorillas, baboons, etc.).Even if "species-level direction of manual lateralization was largely uniform among non-human primates and neither correlated with phylogeny nor with any of the selected biological predictors" according to the quantitative phylogenetic methods used by the authors, such a consistent and significant direction of the effect is not at the species level but at the clade levels across those two different clades raises some questions (if not two consecutive false positive or driven by cofounding factors other than ecology). Given those differences are driven by some overrepresented species, does it mean that effect is significant only when considering species with high sample size and robust statistical power? Does it then mean that lack of effect at the species level might be attributed to low sample sizes in most of the species? Regarding the highly heterogenous sample size across species and effects found at the clade level, the authors should address those questions and such a statistical power issue which could make the results at the species-level equivocal regarding the hypothesis tested in this study.

Thank you for sharing these findings with us. Since we have adjusted the coding of some PGLS predictors (see below) we now find a weak ecological signal in the direction data and discuss it in the manuscript (especially l. 479 ff.) However, we are cautious about the tests presented in your review, because they were run with individual-level data (thus being sensitive to sampling biases) and do not address phylogeny.

By concentrating on species level means, we tried to circumvent some critical sample-size related biases in our analyses and are convinced that our PGLS approach (while being aware of its limitations) is better suited to explore the patterns discussed. Unfortunately, we lack a sufficient number of species (~20) to run our models at clade level. However, we certainly agree that it would be very interesting to have such analyses in future studies. We also share your opinion that the scarcity of positive findings in the one-sample t-tests surely is related to sample size issues. This problem is discussed in lines 429 ff.

(2) Lines 268-269: "To prevent these taxa to skew the comparisons, we restricted their respective sample sizes to the numbers of individuals in the second-largest species sample for their respective clade."To be honest, although I understand this typical issue (that highly represented species naturally skew the results), the issue could be seen the other way around (as mentioned in the previous comment 1): Considering the hypothesis that a high sample size is critical at the species-level to evaluate and infer population-level handedness in nonhuman primates, it is particularly unfair and strange to arbitrary restricted the sample-size and weaken the only data-sets that are relevant and significant because of their statistical power. After such a generalization of the weaknesses of the sample sizes even to the species with strong evidence, the lack of effects of the selected biological predictors at the species-level as well as the uniformity of the species-level direction of manual lateralization reported among non-human primates remains then questionable.If the analyses are made from the individual HI scores, how such a restriction has been done? Which individual HI scores in the chimpanzees and in the olive baboons have been kept in the analyses and which ones have been excluded?

First, there might be a misunderstanding here: "To prevent these taxa to skew the comparisons, we restricted their respective sample sizes to the numbers of individuals in the second-largest species sample for their respective clade." This step was only taken to compute a clade-specific baseline distribution of right/left/ambipreferent individuals to compare species-level distributions by means of a Chi^2^ test. It was therefore in our interest to make these numbers representative for the clades and not particular species. We did not exclude individual HI scores or made arbitrary restrictions to the results.

“the lack of effects of the selected biological predictors at the species-level as well as the uniformity of the species-level direction of manual lateralization reported among non-human primates remains then questionable.”

This aforementioned procedure was only done in preparation for the Chi^2^ tests, the aspects you mention here were addressed by the PGLS models. These two analyses were performed independently from one another.

“If the analyses are made from the individual HI scores, how such a restriction has been done?” As Chi^2^ tests were concerned here, we did not analyze HI scores but the frequencies of individuals being classified as either right/left/ambipreferent and thus worked with categorical data. Therefore, a restricted sample could easily by adjusted to represent the same distribution as the original one (same relative representation of hand preference categories among the species sample).

Second, however, we now checked whether the sample size restriction actually produced different outcomes compared to analyses with the full sample and the deviations are minimal. We therefore decided to remove the sample size restriction from the manuscript.

(3) How terrestrial versus arboreal lifestyle has been attributed to species? Especially macaques? Nemestrina are usually classified as terrestrial while fascicularis and Silenus as arboreal (sylvanus and Tonkeana as both terrestrial and arboreal)?Some species seem to have been attributed to the wrong category. For instance, *Cercocebus torquatus*, Cercopithecus diana, *Cercopithecus neglectus* are usually classified as mainly arboreal (although some of them could also move on the ground) but could not be considered as mainly terrestrial.

We agree that the binary categorization into arboreal and terrestrial species is at best a rough estimation, but for our analysis it appeared to be the most sensible step (we now mention this as an additional caveat to the validity of our findings, see l. 552 ff). We originally adapted the ecological classification by Powell et al. (2017) as made transparent in Supplementary Table 4. Regarding your examples: We did in fact classify *C. diana* as an arboreal species, while *C. neglectus* and the collared mangabey were initially considered terrestrial here. Within the ecological spectrum of primates, we would argue that *Cercocebus torquatus* indeed displays a strong terrestrial bias, which is both evident from morphology and observations from the wild suggesting that adults spend more than 50% of their waking time on the ground (Nakatsukasa, 1996; Cooke, 2012). We are thus comfortable with grouping *Cercocebus* with the terrestrial taxa. After reviewing the literature on habitat use in *C. neglectus* more thoroughly, we agree that an arboreal categorization is more fitting for this species (while it mostly dwells within the lower forest strata it only spends about 20% of its time on the ground – Gautier-Hion, 1988). We also adapted the ecological classification for *Macaca nemestrina* (now listed as terrestrial, since more time is spent on the ground than in trees (Ruppert et al., 2018)) and we no longer include bonobos as habitual tool users in the wild, since they lack foraging related/extractive tool use that is of special relevance for the tool use hypothesis of handedness evolution.

(4) Lines 483-484: "it is obvious that other hand use situations, such as gesturing, need to be considered to arrive at a holistic understanding of primate hand use evolution"Do you mean "communicative gesturing"? If so, a reference might be useful here.

We have now specified this phrase and included references on the topic.

(5) I'll be happy to share our baboon *Papio anubis* data set (Vauclair et al. 2005) from which the HI scores have been not included in the meta-analysis.

Thank you so much for providing this valuable dataset! It markedly improved our study sample.

References:

Nakatsukasa, M. (1996). Locomotor differentiation and different skeletal morphologies in mangabeys (*Lophocebus* and *Cercocebus*). Folia Primatologica, 66(1-4), 15-24.

Cooke, C. A. (2012). The feeding, ranging, and positional behaviors of *Cercocebus torquatus*, the redcapped mangabey, in Sette Cama Gabon: A phylogenetic perspective. The Ohio State University.

Powell, L. E., Isler, K., and Barton, R. A. (2017). Re-evaluating the link between brain size and behavioural ecology in primates. Proceedings of the Royal Society B: Biological Sciences, 284(1865), 20171765.

Gautier-Hion, A (1988) Polyspecific associations among forest guenons: Ecological, behavioral and evolutionary aspects. In A Gautier-Hion, F Bourliere, J-P Gautier, and J Kingdon (eds.): A Primate

Radiation-Evolutionary Biology of the African Guenons. Cambridge:Cambridge Press, pp. 452-476.

Ruppert, N., Holzner, A., See, K. W., Gisbrecht, A., and Beck, A. (2018). Activity budgets and habitat use of wild southern pig-tailed macaques (*Macaca nemestrina*) in oil palm plantation and forest. International Journal of Primatology, 39(2), 237-251.

[Editors’ note: further revisions were suggested prior to acceptance, as described below.]

1. Reviewer 2 makes an argument that the paper is overly dismissive of the (novel) POH hypothesis, given weak evidence for a potential role of terrestrial versus arboreal ecology in lateralization direction (in the predicted direction). The reviewer also asks for more discussion of whether sample size/power issues contribute to weak support for existing hypotheses.Please consider whether you would like to moderate language in your abstract or discussion in light of these arguments (the Reviewer points to some specific examples). Ultimately, the interpretation of your results is not extreme and is your prerogative, so we do not consider this a prerequisite for publication. However, it may be premature to base support or rejection of a long-standing hypothesis based on p-values of 0.04 versus 0.07, which have a very similar level of evidentiary strength (or weakness).

Thank you for these clarifications. In parts, there has been a misunderstanding which we hope to have cleared up in the current version of the text. We reject the theoretical framework of the postural origins hypothesis but not the possibility of ecological effects on lateralization direction – on the contrary, we openly communicate that these effects are detectable in our sample (l. 193 f). We adjusted various phrasings throughout the text to reflect that. We now also stress the limitations of our finding that ecology affects hand preference strength more than in earlier versions of the manuscript.

While screening our submission again, we noticed several issues with the raw data which we have now fixed. These concerned the assignment of individuals into handedness categories (due to a copying error numerous subjects had previously been misclassified, affecting results of the Chi^2^-tests) and typo for the HI value of a single *S. apella* individual (Vincent: HI = – 1 instead of 0.1). This forced us to rerun all of our phylogenetic analysis. The results are essentially unchanged, but certain decimal places had to be adjusted in the respective tables.

Reviewer #2 (Recommendations for the authors):Thank you for the revision of your manuscript. I am glad that the authors finally acknowledged and reported a significant effect for the ecological signal (arboreal versus terrestrial ecology) in the direction data, which, to me, change significantly the initial straightforward conclusions of the paper. But I am afraid that the authors preferred to maintain their initial firm and unbalanced conclusion after downplaying the potential relevance of this effect. I am also satisfied by the responses and justifications provided in the comments 2 and 3.

Thank you for this second evaluation of the manuscript. We cannot concur with your opinion that the factor ecology in explaining lateralization direction had been downplayed in our work. Instead, we would argue that it is discussed quite extensively. In the current version of the article, we have further elaborated on this issue and are convinced that it adequately acknowledges the possible significance of ecology on lateralization direction.

Here are my remaining concerns.1) Ecological factor (arboreal versus terrestrial ecology) is presented in the paper as one of the few key biological factors and hypothesis that are willing to be tested in the present study. It turned out, that, in the present revision after my first round of comments, the authors finally did find a significant result (p=0.04) for this ecological signal in the direction data (which still reaches conventional level of significance at p<0.07 even after removing the human species). To be honest, I found thus quite surprising, partial and unfair that this result is poorly discussed, not even mentioned in the abstract but rather even just quickly downplayed in the discussion with any justification, other than "the weakness of the effect" and other than a laconic claim (at lines 307-311) : "Admittedly, improved sampling, both within and across species might consolidate this pattern in the future, so that we do not want to dismiss the POH at this point. However, even if its predictions should become validated eventually, the hypothesis does still not provide an evolutionary mechanism for why population-level hand preference patterns should be coupled to positional behavior."This is unfortunate and a bit short as a discussion after such a flipping finding that might be consistent with the "novel PO hypothesis". Despite evidence, the authors preferred ignoring interpreting this potential effect (other than downplaying it) and give thus the feeling of not wanting to contradict their initial firm conclusion and thus kept only one side of the potential interpretations, namely for instance:– "Species-level direction of manual lateralization was largely uniform among non-human primates and did not notably correlate with any of the selected biological predictors" (lines 19-21).

In our opinion, less ambiguity in the data would be required to confidently speak of a “notable” correlation or a “flipping finding” and we hope that the reviewer respects our disagreement here. We now changed “notably” to “strongly” for greater clarity. We would very much prefer to not discuss this point further in the abstract, though (we have already reached the word limit). Ecological effects on hand preference strength are emphasized, since this finding is both significant and novel.

– "Furthermore, the evolutionary scenario proposed by both versions of the POH is outdated and should not be perpetuated without explicitly stating its shortcomings." (lines 313-314)

We, in fact, want to defend this point and see now new arguments provided to adjust it here. Even if we would have recovered results clearly matching the predictions of the POH, the proposed evolutionary scenario underlying it would still not hold up and require adaptation, as we now describe in more detail in the text (l. 322 f). One point probably deserves further emphasis to avoid misunderstandings: We reject the theoretical framework of the postural origins hypothesis but not the possibility of ecological effects on lateralization direction – in the contrary, we openly communicate that these effects are detectable in our sample (see below) and that future studies may well demonstrate their robustness (l. 319).

– "no phylogenetic or ecological signal was found for lateralization direction, and few species exhibit significant population-level hand preferences. We tested (…) the POH (…) hypotheses, but none were corroborated by our data." (lines 435-438)

We adjusted the text to “In contrast to this, no phylogenetic signal and weak ecological effects were found for lateralization direction, and few species exhibit significant population-level hand preferences.”.

We indeed do not consider our results to corroborate the POH, as noted above.

I totally agree that given (1) the effect is not strong, not at p<0.0001, but rather at 0.04 and (2) even decreased at p=0.07 when excluding humans, and that (3) we should indeed never exclude, in behavioral science, a potential false positive effect, (4) and that further investigation would help keep testing this small effect by increasing statistical power for each species. So any firm conclusion should not be drawn from such a small effect, let alone the firm conclusion of the authors ignoring this effect. Nevertheless, I recommend the authors to, at least, try to interpret this potential effect which seems congruent with the "novel PO hypothesis" by presenting this other side of the potential interpretations, not just one side that does fit with their initial conclusion.A more balanced view including both opposite sides of the interpretation of this effect would be thus more fair and impartial (i.e. "congruent with the "novel PO hypothesis" versus "congruent with the initial conclusion of the paper").

We are surprised that the reviewer got this impression from the text as we clearly state that “whereas habitual tool use and absolute brain size clearly do not influence the direction of lateralization among anthropoids in general, the analyses provide evidence for a weak but detectable effect of ecology” (l. 193 f). Given our data we do certainly not downplay this potential effect and (as mentioned above) we dedicate a significant portion of the Discussion section to the POH.

Moreover, when I played with the data analyses (and likely with the grey zone of attribution of "arboreal" versus "terrestrial" categorial attribution for some ambiguous species, see previous initial comment 2 and its response), I found a stronger and significant difference between the mean HIs of terrestrial versus arboreal non-humans (terrestrial : M.HI = 0.11, SE = 0.02; arboreal: M.HI = -0.06, SE = 0.03) in comparison to the authors with their justified classification (MeanHIarboreal: -0.08, SD: 0.16; MeanHIterrestrial: 0.04, SD: 0.10).

We fully agree that there is ambiguity in the ecological classification of the tested species. We tried to make this transparent in the Discussion section: “Finally, we need to acknowledge the limitations of our phylogenetic modelling approach. In particular, the binary coding of ecology (arboreal vs terrestrial) obviously simplifies the remarkable spectrum of positional behaviors found among anthropoids. Future approaches might explore alternative strategies to statistically code this multifaceted variable” (l. 392 f). Yet, we want to stress that group-level means should only be evaluated statistically when taking phylogenetic dependencies into account.

Bonus comment (for the pleasure of the discussion):Maybe one evolutionary hypothesis to try interpreting the effect in accordance with the "novel PO" might be that, in the primate evolution, a tendency to predominance of right-handedness would be inherited from a the common ancestor between Catarrhini and Hominid (but not from the Platyrrhini). In that case, this right-bias would be thus mostly altered in arboreal ecology (given its implication in supporting the body in the trees) whereas more visible in terrestrial ecology without such a body postural constraint; a right predominance bias that would dramatically increase in the course of the evolution in the bidedal *Homo sapiens*, the most terrestrial primates (but also to a lesser extend the gorillas, the second most terrestrial primates).So, as mentioned by the authors, only further investigations by increasing the sample size per species (and the number of species) would help disambiguating those two opposite sides, but for that, the two sides should be fairly considered in the paper…

Thank you for sharing this idea with us. As you point out, more data would certainly be required to test its validity. For the time being, however, we would like to point to the result of our own ancestral character modelling, which does not suggest a noteworthy right-handed tendency in the catarrhine (HI: -0.018), cercopithecoid (HI = -0.05) or hominoid (HI = 0.02) common ancestor (while keeping the noise in the estimates in mind).

2) Lines 255-257: "It is therefore premature to assume that hominids display qualitatively different population-level lateralization patterns than other primates."Lines 268-270: "Again, we suggest that an expanded and more balanced dataset could level out the lateralization differences between (great) apes and other anthropoids."I am not sure to understand why? Does it mean that increasing the statistical power in other non-hominid species (such as the significantly right-handed baboon *Papio anubis*, at N = 104) to reach the sample size of great apes would likely make them converge toward more significant biases at the populational-level? If so, it would be very interesting for getting the full picture of primate handedness evolution.Or in contrary, decrease the statistical power in great apes would make them non-significantly lateralized (but less representative), just like most of the non-hominid primates?In both cases, the rational of those both alternatives are not entirely clear to me.I would rather suggest the authors to provide, as I initially suggested in my first review, a proper and full discussion about the issue of sample size that occurred in the primate handedness literature, and its minimum threshold to be representative enough for inferring and comparing direction and degree of population-level handedness among nonhuman primates.In fact, given the weaker degree of population-level handedness bias in nonhuman primates in comparison to humans, increasing sample size of subjects in nonhuman primates is thus an important consideration for being able to detect representative directional and degree of asymmetries at population-level and requires thus the largest sample size of subjects whenever possible (see the statistical demonstration in Hopkins and Cantalupo, 2005, and see Hopkins 2006, recommending more than 50 subjects per species).Only 5 species in the literature reviewed in the present paper reached that sample-size threshold of 50 subjects and that level of statistical power, and actually…all of them showed significant population-level handedness!Gorillas, Gorilla gorilla, N = 76 => significant right-handednessAdult Bonobos, Pan Paniscus, N = 52 adults => significant right-handednessChimpanzees, *Pan troglodytes*, N = 536 => significant right-handednessBaboons, *Papio anubis*, N = 104 => significant right-handednessHumans, *Homo sapiens*, N = 127 => significant right-handednessEven after downing the sample-size threshold to 40 subjects, out of the 7 species which reached that statistical power, only 1 species, the *Ateles* fuscicepsall (N=46), does not show any populational bias, while the Orangutans, Pongo sp., N = 47 => significant left-handedness.So one of the main limitations of the nonhuman primate literature (including the present study) on both handedness and brain asymmetries is the lack of sample size which prevent doing robust comparative analyses at the species-level with humans or great apes. With such a lack of statistical power in 33 species (out of 38) which are below this sample size threshold, it should not be excluded that we may have to deal with not representative and inconsistent findings (e.g., strong bias in some monkeys, lack of populational bias in other, etc.), which could not be completely covered and addressed by clade-level analyses. This is a discussion point that should be better addressed.

Please note, that this sentence refers to age group sampling. Nevertheless, we adjusted it for greater clarity.

In general, we fully agree with your point that increasing the samples of non-hominids would be greatly beneficial for comparative handedness research. We also directly express this in the text on various occasions (e.g., l. 270 f, l. 278 f, l. 458) und would prefer from repeating this point once again in the manuscript. However, establishing and/or discussing a minimum sample size that might be utilized for such approaches goes beyond the scope of the paper and we are unaware of any sound mathematical modelling to infer such a threshold. In fact, we are unsure whether it could be a meaningful mark. The observation that well-represented species frequently exhibit significant population-level handedness must not be overestimated: Since the one-sample t-test becomes very sensitive at large sample sizes, such significant findings may even be rather trivial and overshadow diversity in hand preference patterns among both significantly handed taxa and those that are not. For instance, both chimpanzees and humans show undoubtedly population-level handedness in the tube task or comparable paradigms. Still, the distribution of their hand preferences is markedly different and should not be equated. We would argue that chimpanzee hand preferences have more in common with species that lack population-level handedness than with humans. Just achieving statistical support for population-level handedness is, in our opinion, not a particularly meaningful goal in comparative contexts. We added a sentence to emphasize this point (l. 260 f).

We also want to stress that aside from the baboons (which at the genus level do not score population-level handedness here, *p* = 0.08 – see Tab. 1) all of the species you mention are hominids and thus represent just a single lineage which might not be representative for anthropoids as a whole. In fact, we sampled several genera with highly uniform species (both ecologically and morphologically) that exceed the suggested n = 50 threshold, without showing evidence of population-level lateralization: *Leontopithecus*, *Ateles*, *Sapajus*, *Hylobates* (and *Papio*). In *Cercopithecus* (n = 45) we also do not even see a trend.

3) At table 1, the data and analysis presented from our baboons' study *Papio anubis* (N=104), are not accurate and might affect the general analyses of the paper. Only 84 subjects are mentioned, instead of N = 104, and the value of the M.HI, presented inaccurately as non-significant (p=0.102), does not correspond. In our results, the M.HI was significant (p<0.05) and was rather M.HI = 0.13, SE = 0.06, N=104 (see the updated data file that I sent including the *Papio anubis* data).

According to our inclusion criteria, we only considered subjects that scored at least 30 insertions, thus we had to exclude 20 baboons that did not fit this criterion (see Materials and Methods).

Moreover, this study is the only one in monkeys approaching the sample size and statistical power of most great apes' studies, converging with significant populational bias finding. I believe this finding should be better integrated within the discussion made by the authors about significant findings in large sample size in great apes only, as well as the discussion about the "sample size issues in handedness NH primate studies" requested in the previous comment.

We do not aim to evaluate findings made by other studies in the Discussion section of our manuscript. In fact, we yielded different results for the genus *Papio* (n = 108) in our study, when applying stricter criteria of subject inclusion (min. number of insertions = 30). Also, we preset genus-level data on several other monkey lineages that exceed the sample size available for some great ape genera without them showing significant population-level handedness (*Ateles*, *Sapajus*, *Leontopithecus*).

Overall, despite my concerns, I insist that this study is very important for the field (that is why I believe the conclusions should be thus more balanced) and I renew my congratulations on this amazing and considerable work. I had such a pleasure reading it and hope my comments and questions were useful.

Thank you so much for this positive evaluation! We very much appreciated the detailed critique and discussion.